



**Transboundary water sharing policies conditioned on hydrologic variability to**
**inform reservoir operations**
Guang Yang[1], Paul Block[2]
**Abstract** Water resources infrastructure is critical for energy and food security, however, the development
of large-scale infrastructure, such as hydropower dams, may significantly alter downstream flows,
potentially leading to water resources management conflicts and disputes, especially in transboundary
river basins. Mutually agreed upon water sharing policies for the operation of existing or new reservoirs
is one of the most effective strategies to mitigate conflict, yet this is a complex task involving the
estimation of available water, identification of users and demands, procedures for water sharing, etc. We
propose a water-sharing policy framework that incorporates reservoir operating rules optimization based
on conflicting uses and natural hydrologic variability, specifically tailored to drought conditions. We first
establish the trade-off between downstream and upstream water availability utilizing multi-objective
optimization of reservoir operating rules. Next, we simulate reservoir operation with the candidate
(optimal) rules, evaluate their performance, and select the most suitable rules for balancing water uses.
Subsequently, we build a relationship between the reservoir operations simulated from the selected rules
and drought-specific conditions to derive water-sharing policies. Finally, we re-optimize the reservoir

[1] Department of Civil and Environmental Engineering, University of Wisconsin-Madison, 1415 Engineering Dr.,
Madison, WI 53706. E-mail: gyang82@wisc.edu

[2] Department of Civil and Environmental Engineering, University of Wisconsin-Madison, 1415 Engineering Dr.,
Madison, WI 53706 (corresponding author). E-mail: pblock2@wisc.edu



operating rules to evaluate the effectiveness of the drought-specific water sharing policies. We apply the
framework to reservoir operation of the Grand Ethiopian Renaissance Dam (GERD) on the Blue Nile
River. We find that the derived water sharing policy can balance GERD power generation and downstream
releases, especially in dry conditions, effectively sharing the hydrologic risk in inflow variability among
riparian countries. The proposed framework offers a robust approach to inform water sharing policies for
sustainable management of transboundary water resources.
**Keywords:** Reservoir operation; Water sharing policy; Drought mitigation; Multi-objective optimization;
Grand Ethiopian Renaissance Dam.

## 1. Introduction

Rapid population growth and socio-economic development exacerbate stress on the management of
water resources globally (Vörösmarty et al., 2000;WWAP, 2019). Surface-water reservoirs and their
effective management is one of the most efficient means to reduce this stress by reallocating water
resources spatially and temporally (Gaudard et al., 2014). Thus in recent decades, many models and
strategies have been investigated to inform and improve reservoir operation decision-making (Lerma et
al., 2015;Chaves and Chang, 2008;Cancelliere et al., 2002;Giuliani et al., 2015a;Herman and Giuliani,
2018;Karamouz and Houck, 1982;Consoli et al., 2008;Giuliani et al., 2014;Oliveira and Loucks, 1997).
In general, reservoir decisions (e.g., water releases and power generation) are determined using reservoir
operating rules with available input variables including reservoir state (e.g., reservoir water level) and
hydrological conditions (e.g., reservoir inflow) (Oliveira and Loucks, 1997).
Reservoir operating rules are typically derived using fitting-based and simulation-optimization-based
approaches. In fitting-based rules derivation, reservoir operation decisions are optimized and subsequently



fitted with input variables using linear regression (Bhaskar and Whitlatch Jr, 1980), artificial neural
networks (Cancelliere et al., 2002), fuzzy inference (Chang and Chang, 2001), and decision trees (Wei
and Hsu, 2008). For example, Karamouz and Houck (1982) developed annual and monthly reservoir
operating rules by regressing reservoir releases optimized from deterministic dynamic programing onto
reservoir decision-making state variables. Cancelliere et al. (2002) derived the operating rules of an
irrigation supply reservoir by using neural networks techniques and found that the rules can improve
reservoir operation performance during drought conditions. Goyal et al. (2013) compared the performance
of artificial neural networks, fuzzy logic, and decision tree algorithms for deriving the operating rules of
an irrigation and power supply reservoir in northern India. In simulation-optimization methods, the
parameters of reservoir operating rules are optimized with an iterative simulation-based search algorithm
in which the performance is evaluated directly from reservoir operation simulations (Le Ngo et al.,
2007;Rani and Moreira, 2010). For example, Giuliani et al. (2015a) approximated reservoir operating
rules by using artificial neural networks and radial basis functions (RBFs) and optimized the rules for
multi-purpose water reservoirs with an evolutionary algorithm.
Most of these approaches are implemented in water resources systems contained within a basin or
jurisdiction in which the benefits (e.g. power generation, water supply, and ecosystem function
maintenance) can be quantified and evaluated (Reddy and Nagesh Kumar, 2007;Feng et al., 2018;Yang
et al., 2016). Reservoir operations in transboundary river basins are necessarily more complex given a
wide variety of social, political, economic, cultural, and physiographic conditions (Zeitoun and Mirumachi,
2008). Disputes and conflicts are not uncommon between riparian states in transboundary river basins
when water sharing agreements are non-existent or non-enforceable and claims may be defined based on
historical use. For example, the Nile River serves eleven countries, 250 million people (Nile Basin
Initiative, 2017), and is vital to agriculture, industry, and hydropower, (Paisley and Henshaw, 2013), yet


no mutually agreed upon water sharing policies exist. (The 1959 agreement (Guariso and Whittington,
1987) has been repudiated by upstream riparian states.) Acknowledging significantly divergent interests
and a history of conflict and distrust, quantifying the impact of reservoir operation on downstream benefits
is challenging, hindering development of water sharing strategies (Link et al., 2016).
According to the Transboundary Freshwater Dispute Database (McCracken and Wolf, 2019), the
existing 310 international river basins across the world are shared by 150 countries and disputed areas,
cover 45% of the Earth's land surface, and contribute to 60% of the world's freshwater resources. As
suggested by Sadoff and Grey (2002), it is critical to understand and account for the range of inter-related
benefits resulting from the development of international rivers in a cooperative way. Such cooperation of
water resources development in international river basins has been widely investigated in recent years (Li
et al., 2019;Luchner et al., 2019;Anghileri et al., 2013;Uitto and Duda, 2002;Dombrowsky, 2009;Tilmant
and Kinzelbach, 2012;Arjoon et al., 2016;Wu and Whittington, 2006;Wheeler et al., 2018;Goor et al.,
2010;Degefu et al., 2016). For example, Arjoon et al. (2016) proposed a benefit-sharing method based on
the optimization results from a hydro-economic model and evaluated the value of cooperative water
management in the Eastern Nile River basin; Li et al. (2019) analyzed the water benefits of stakeholders
from transboundary cooperation under different reservoir operation scenarios by using cooperative game
theory methods; Luchner et al. (2019) simulated reservoir operations and water allocation in an
international river basin in Central Asia and found that international cooperation in the power sector can
ease the conflict between upstream hydropower production and downstream irrigated agriculture.
Although cooperation in transboundary river basins can result in a win-win situation for both
downstream and upstream stakeholders, cooperative water use strategies are obstructed by single-sector
interests, especially when long-term commitments are involved (Wu and Whittington, 2006). More



specifically, it is often difficult to achieve a mutually agreed-on cooperation strategy given divergent
solution preferences by stakeholders. Additionally, there is no standard that regulates how the benefits of
water use from various sectors (e.g., drinking, agriculture, industry, recreation, and navigation) are
quantified and what mechanism should be used to allocate/share the benefits (Arjoon et al., 2016;Acharya
et al., 2020). Thus, most previous studies focus on evaluating the impact of cooperative operation in
transboundary river basins and illustrating the importance of a cooperative strategy through water system
optimization and simulation (Goor et al., 2010;Anghileri et al., 2013;Uitto and Duda, 2002;Dombrowsky,
2009;Tilmant and Kinzelbach, 2012;Luchner et al., 2019). There is less literature, however, addressing
strategies for reaching an agreement or consensus on water resources development amongst downstream
and upstream riparian countries in transboundary river basins (Wheeler et al., 2016;Li et al., 2019;Degefu
et al., 2016).

In this study, we propose a systemic framework to derive operational reservoir water-sharing policies

using multi-objective optimization for water use conflict mitigation. Specifically, we (1) optimize
reservoir operating rules and establish trade-off between downstream and upstream water availability, (2)
simulate reservoir operation with the candidate (optimal) rules, evaluate performance, and select the most
suitable rules for balancing benefits, (3) derive water-sharing policies conditioned on reservoir operations
and water availability forecasts, and (4) re-optimize reservoir operating rules incorporating derived water-
sharing policies to evaluate effectiveness and performance. We select the Grand Ethiopian Renaissance
Dam (GERD) in Ethiopia to demonstrate the framework and illustrate how operational water-sharing
strategies, reflective of upstream and downstream demands and natural hydrologic variability, can
promote water-sharing agreements between upstream and downstream countries.



## 2. Study Area and Data

### 2.1. The Blue Nile Basin and the Grand Ethiopian Renaissance Dam

The Blue Nile River, the largest tributary of the Nile River, originates at Tana Lake in Ethiopia and merges with the White Nile River in Khartoum, Sudan. Average annual rainfall in the upper part of the basin varies between 1200 and 1800 mm (Conway, 2000), with a dominant rainy season in June–September contributing approximately 70% of mean annual precipitation. During this season, the Blue Nile contributes nearly 80% of the total Nile River streamflow (Swain, 2011) and the average annual runoff of the Blue Nile at Roseries, near the Ethiopia–Sudan border, is approximately 49 km$^3$ (Wheeler et al., 2016), thus it plays a significant role in livelihood and development in Ethiopia, Sudan and Egypt.

Ethiopia started constructing the Grand Ethiopian Renaissance Dam (GERD) across the Blue Nile River in 2011, approximately 15 km upstream of the Sudanese border (Fig. 1). When completed, the GERD will become the largest hydroelectric dam (installed capacity exceeding 5,000MW) in Africa (Government of Ethiopia, 2020) and will have a total reservoir capacity of 74 billion cubic meters. The GERD is expected to produce an average of 15,130 GWh of electricity annually (Tan et al., 2017), which will contribute to Ethiopia's national energy grid and feed the East African power pool (Nile Basin Initiative, 2012). Although the GERD is primarily designed for hydropower, and thus non-consumptive, operating to maximize power generation may result in a water release schedule significantly different from the natural annual cycle, particularly considering drought periods, with implications to Sudan and Egypt. This is the crux of the current hydro-political confronting the riparian countries.

In this study, we develop GERD reservoir operation rules considering power generation and downstream water release simultaneously to mitigate upstream-downstream water use conflicts, particularly tailored to drought periods. The study investigates water-sharing (drought mitigation) policy




derivation procedures (reservoir operation simulation and optimization, power generation and
downstream water release analysis, drought mitigation policy extraction and validation) balancing GERD
production and downstream flow volumes. Historical monthly inflow at El Diem gauging station (located
just downstream of the GERD site) for 1965-2017 (Fig. 2) are applied.

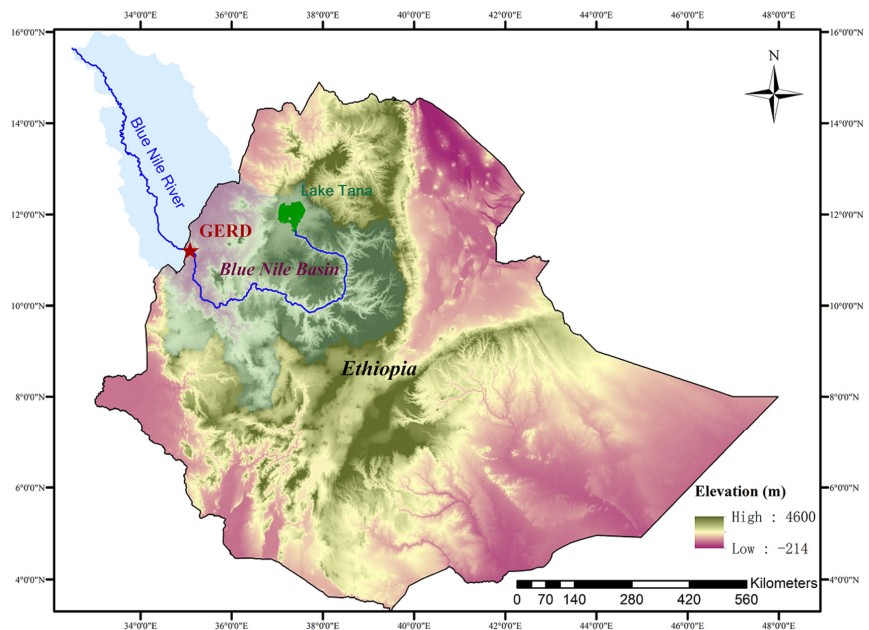


Fig. 1 Blue Nile basin with Ethiopia country borders and the location of the GERD reservoir



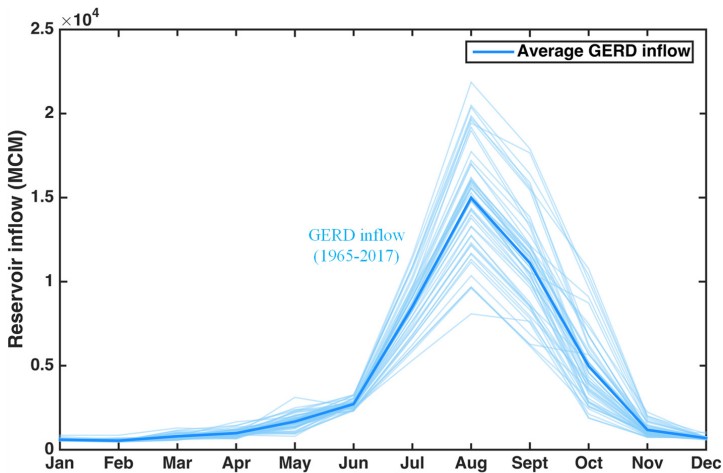


Fig. 2 Monthly inflow into the GERD reservoir during 1965-2017.

## 3. Models and Methods


The procedure for water sharing policy derivation and evaluation for transboundary rivers including
large-scale reservoir operations is introduced in this section (Fig. 3). In summary, the process is as follows:
• First: optimize the reservoir operating rules approximated with Radial Basis Functions (RBFs)

and obtain the Pareto front for upstream and downstream benefits trade-off.

• Second: select feasible solutions on the Pareto front according to the requirements of power

generation and drought mitigation; specifically, for a given power generation level, the

distribution of annual water release is analyzed with special attention to low flow years.

• Third: define the relationship between annual reservoir inflow and releases based on the selected

feasible solutions; the policy can be further framed as a function of reservoir annual inflow

predictions.

• Fourth: incorporate the policy into general RBFs-based rules to evaluate policy effectiveness and

robustness.

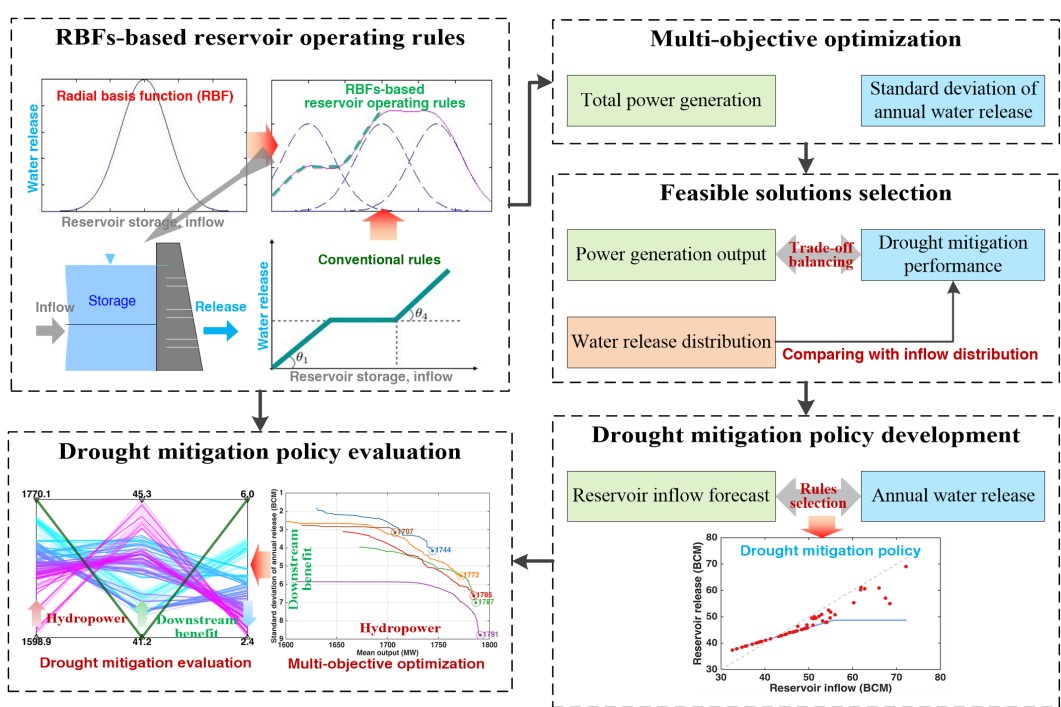

Fig. 3 Procedure of drought mitigation policy derivation and evaluation for reservoir operation in transboundary rivers.
3.1. **Reservoir operation model**

The primary purpose of the GERD reservoir is hydropower production; this objective function can

be described as follows:

$$Max \quad E = \sum_{t=1}^{T} P_t \cdot \Delta t, \quad P_t = \eta \cdot g \cdot \rho \cdot Q_t^P \cdot H_t^P / 1000 \qquad (1)$$

where $E$ is hydroelectricity generation (kW h) during total number of operational periods $T$; $P_t$ is the
power generation output (kW) during time period $t$ and $\Delta t$ is the time (h) of a single period; $\eta$, $g$, and $\rho$



refer to the dimensionless hydropower generation efficiency of the turbines (set as 0.9 in this study),
gravitational acceleration (9.8 m/s$^2$), and water density (1000 kg/m$^3$), respectively; and $Q_t^P$ and $H_t^P$ are
reservoir release for power generation (m$^3$/s) and average power head (m) in period $t$, respectively.
In lieu of modeling specific water requirements downstream of the GERD, minimizing annual water
release variance is applied. This approach favors reliable releases yet also reflects natural hydrologic
variability, and can be described as below.
$$Min \quad Std\left(Q_y^{out}\right) = \sqrt{\frac{\sum_{y=1}^{Y}\left(Q_y^{out} - \bar{Q}_y^{out}\right)^2}{Y}} \qquad (2)$$

where $Q_y^{out}$ is the reservoir water release in year $y$ and $\bar{Q}_y^{out}$ and $Std\left(Q_y^{out}\right)$ are the mean value and standard
deviation of reservoir annual water release across all operational years $Y$, respectively.
Physical and operational reservoir constraints are listed as below.
(a) Water balance:
$$S_{t+1} = S_t + \left(Q_t^{in} - Q_t^{out}\right) \cdot \Delta t - EP_t \qquad (3)$$

where $S_t$ and $S_{t+1}$ are reservoir storage (m$^3$) in period $t$ and $t+1$, respectively, $Q_t^{in}$ represents reservoir
inflow (m$^3$/s) in period $t$, $Q_t^{out}$ is reservoir release (m$^3$/s) in period $t$, and $EP_t$ is the sum of evaporation and
seepage from the reservoir (m$^3$) in period $t$.
(b) Reservoir capacity limits (Jameel, 2014):





The reservoir structural and operational constraints can be expressed as:
$$S^{\min} \le S_t \le S^{\max} \qquad (4)$$

where $S^{\min}$ and $S^{\max}$ are the minimum and maximum allowable reservoir storage (m³), respectively.
Additionally, $S^{begin}$ and $S^{end}$ represent the initial and final reservoir storage (m³) for simulations,
respectively, and are prescribed as:
$$S_t = \begin{cases} S^{begin} & t = 1 \\ S^{end} & t = T \end{cases} \qquad (5)$$

(c) Reservoir release limits:
The reservoir release constraints are expressed as:
$$QL_t \le Q_t^{out} \le QU_t \qquad (6)$$

where $QL_t$ and $QU_t$ are the minimum and maximum release (m³/s) in period $t$, respectively. The
expected guidelines for GERD reservoir water release are not explicitly available, thus releases are set
lower than the maximum reservoir inflow during the high-flow season to reduce or eliminate downstream
floods.
(d) Power generation limits (Tesfa, 2013):
$$PL_t \le P_t \le PU_t \qquad (7)$$

where $PL_t$ and $PU_t$ are the minimum and maximum power limits (kW) in period $t$, respectively.





3.2. **Reservoir operating rules**

In this study, reservoir water releases are conditioned on radial basis functions (RBFs), a non-linear

function approximating method (Deisenroth et al., 2013;Buşoniu et al., 2011;Giuliani et al., 2015b) which
can provide universal approximation (Tikk et al., 2003) and ensure a flexible reservoir operating rules
structure. For more applications of RBF models in reservoir operation see Giuliani et al. (2015a). The
reservoir operating rules are defined as below.
$$Q_t^{out} = \sum_{u=1}^{U} \omega_u \varphi_u (X_t) \quad t=1,...,T \quad 0 \le \omega_u \le 1 \tag{8}$$

$$\varphi_u (X_t) = \exp \left[ -\sum_{j=1}^{M} \frac{\left( (X_t)_j - c_{j,u} \right)^2}{b_u^2} \right] \quad c_{j,u} \in [-1,1], b_u \in (0,1] \tag{9}$$

where $U$ is the number of RBFs, $\varphi(\cdot)$ and $\omega_u$ are the weights of the $u^{th}$ RBF, $M$ is the number of input
variables $X_t$, and $\mathbf{c}_u$ and $b_u$ are the $M$-dimensional center and radius vectors of the $u^{th}$ RBF, respectively.
Because water release generally depends on the reservoir state (Revelle et al., 1969) and inflow, with
intra- and inter-annual variability, we select reservoir storage $S_t$, inflow $Q_t^{in}$, and seasonal information $\tau_t$
(where $\tau_t$ refers to the position of the current period (month) $t$ within a water year) as input variables and
$X_t = (S_t, Q_t^{in}, \tau_t)$.
In this study, the number of RBFs is set to four (as in  Giuliani et al. (2015b)), thus $U$=4 and $M$=3 (three
input variables) in equation (8)-(9) resulting in 20 parameters in the RBFs-based rules. The parameters in
RBFs-based rules are optimized with a simulation-optimization model (Rani and Moreira, 2010), using



the Pareto-Archived Dynamically Dimensioned Search (PA-DDS) evolutionary algorithm which has been
successfully applied to reservoir operating rules optimization (Yang et al., 2020). The procedure of the
PA-DDS algorithm has been described in detail by Asadzadeh and Tolson (2013).
3.3. **Drought-focused water sharing policy**
To ensure downstream water supply, the GERD reservoir will need to be operated under minimum annual
water release constraints. Apart from the RBFs-based rules determining the reservoir water release in each
time step (months), a drought mitigation policy is also adopted to address dry periods. The policy is based
on annual time steps and represented as a linear regression between annual reservoir inflow and water
release. More specifically, the minimum annual reservoir water release can be determined by the following
equation:
$$R_y^{\min} = \alpha \cdot Q_y^{in} + \beta + z \cdot \sigma_d \qquad (10)$$

where $Q_y^{in}$ and $R_y^{\min}$ refer to reservoir inflow and minimum reservoir water release during year $y$,
respectively; $\alpha$ and $\beta$ are regression parameters estimated from simulations containing reservoir inflow
values below the historical average (approximately 49 BCM). An exceedance parameter z is multiplied
by the standard deviation of the regression residuals $\sigma_d$ to vary how conservative the drought mitigation
policy is (see Fig. 4 for a visualization of exceedance parameters). This policy design favors downstream
releases under drought conditions by supplementing what would occur under natural flow conditions,
however as a trade-off, the minimum reservoir release in any year will not exceed the historical average
reservoir inflow (see the horizontal line in Fig. 4).



In this study, the drought mitigation policy is designed with annual streamflow data, however reservoir
operating rules are derived for monthly operation. With reservoir storage in current month $S_m$, reservoir
inflow in current month $Q_m^{in}$, and the predicted reservoir inflow during the rest of the year $Q_{m+1}^{\prime in}$, …, $Q_{12}^{\prime in}$,
the reservoir water release in current month $Q_m^{out}$ and the rest of year $Q_{m+1}^{\prime out}$, …, $Q_{12}^{\prime out}$ can be obtained from
equations (8) and (9), thus the annual reservoir inflow and water release can be estimated as below.
$$Q_y^{\prime in} = \sum_1^m Q_m^{in} + \sum_{m+1}^{12} Q_m^{\prime in} \qquad (11)$$

$$Q_y^{\prime out} = \sum_1^m Q_m^{out} + \sum_{m+1}^{12} Q_m^{\prime out} \qquad (12)$$

The minimum reservoir water release in year y can be estimated from equation (10) as $R_y^{\prime min}$. To ensure
the minimum annual water release obligation is met, if the estimated annual reservoir water release $Q_y^{\prime out}$
is lower than $R_y^{\prime min}$, the release in current month $Q_m^{out}$ will be corrected according to the following:
$$Q_m^{out} = Q_m^{out} + \left( R_y^{\prime min} - Q_y^{\prime out} \right) \times \frac{Q_m^{in}}{\sum_m^{12} Q_m^{\prime in}} \qquad (13)$$

The estimated variables $R_y^{\prime min}$, $Q_y^{\prime out}$, and $\sum_m^{12} Q_m^{\prime in}$ are updated in each time step. In the last month of each
year, the annual reservoir inflow estimation $Q_y^{\prime in}$ and minimum annual water release estimation $R_y^{\prime min}$ will
be equal to $\sum_1^{12} Q_m^{in}$ and $R_y^{min}$, respectively. If $Q_y^{out} < R_y^{min}$, the reservoir water release in the last month $Q_{12}^{out}$
will be corrected as $Q_{12}^{out} + \left( R_y^{min} - Q_y^{out} \right)$ and the $Q_y^{out}$ will be equal to $R_y^{min}$. Thus annual reservoir inflow
$Q_y^{out}$ will be always greater than or equal to the specified minimum reservoir water release $R_y^{min}$. In this





study, both a climatology inflow forecast ($Q_t'^{in}$ set as the long-term average streamflow for that month)
and a perfect inflow forecast ($Q_t'^{in}$ set to observed reservoir inflow $Q_t^{in}$) are used to evaluate the
performance of the drought mitigation policy.

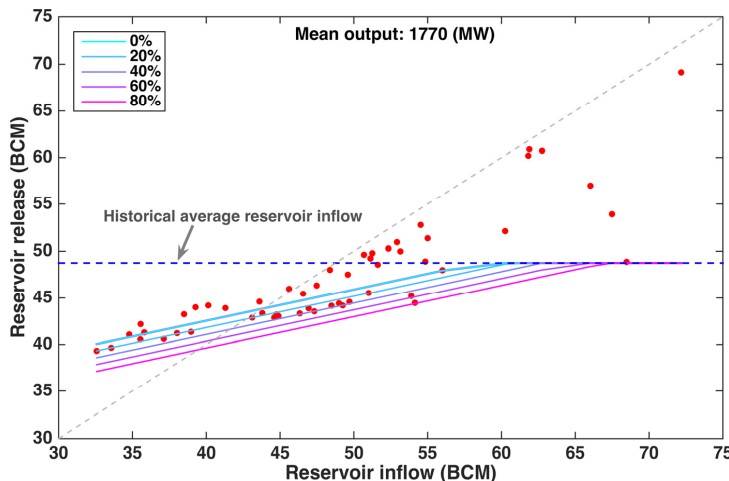


Fig. 4 Sample drought mitigation policy with varying exceedance levels (z=0%, 20%, 40%, 60%, and 80%)

## 4. Results and Discussion


### 4.1. Multi-objective reservoir operation with no drought policy


Multi-objective optimization of GERD reservoir operating rules illustrates that there is a trade-off between
reservoir power generation and deviation in reservoir annual water releases (Fig. 5 (a)). More specifically,
GERD monthly mean power generation output is estimated at 1788, 1708, 1737, and 1707 MW for annual
release standard deviations of 9, 7, 6, 5, and 4 BCM, respectively. Although the reservoir operating rules
are not optimized for maximum annual water release, less $Std\left(Q_y^{out}\right)$ typically leads to relatively more
releases in dry conditions (e.g., 5th, 10th, 15th, and 20th percentiles), especially when the mean output is
greater than 1750 MW (Fig. 5 (b)). Thus downstream countries may benefit more from reservoir operating





rules favoring smaller $Std\left(Q_y^{out}\right)$ in drought conditions; this trade-off between power generation and
$Std\left(Q_y^{out}\right)$ can be used to balance GERD power generation and downstream water use benefits.

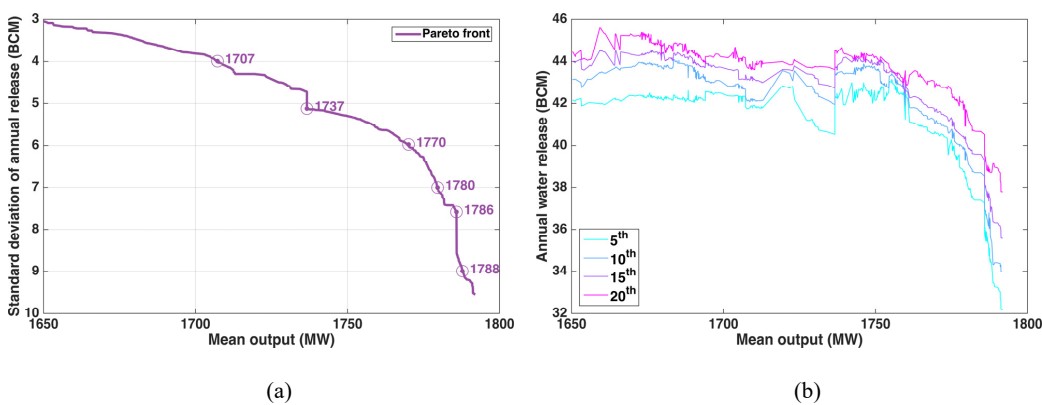

(a)                (b)

Fig. 5 Multi-objective optimization results of reservoir operating rules in terms of (a) Pareto front and (b) the
relationship between power generation and 5th, 10th, 15th, and 20th percentile of annual water release.

The reservoir operating rules simulation results under various mean output levels illustrates that the
variance of annual water release shrinks and reservoir storage declines as power output decreases ( Fig. 6).
Although the median values of annual water release for all six output levels are approximately the same
(around 45 BCM), the reservoir operating rules with more output generally have lower minimum water
releases (Fig. 6 (a) and (b)), especially in dry periods. In general, greater reservoir storage leads to more
power generation (see equation (1)) and vice versa, thus the reservoir operating rules generating 1788 MW
of mean output produces the highest water level, and 1707 MW the lowest (Fig. 6 (d)). Also, there is a
clear trade-off between the variance of reservoir storage and water release (Fig. 6 (a) and (c)); smaller
reservoir storage variance ensures higher reservoir levels, greater water release variance, and lower



minimum water releases. It is worth noting that the 75[th] and 90[th] percentiles of reservoir storage are much
more sensitive to power output than those of water release. More specifically, the 90[th] percentile of water
release for rule types 5 and 6 are almost the same, however, the corresponding percentile of reservoir
storage (as well as power output) are notably different (Fig. 6 (a) and (c)). This indicates that rule type 6
may be inferior to rule type 5 despite of the trade-off in Fig. 5 (a). Thus, it is necessary to analyze the
operation results (including water release and power generation) before selecting the reservoir operating
rules based on the Pareto front in Fig. 5 (a).

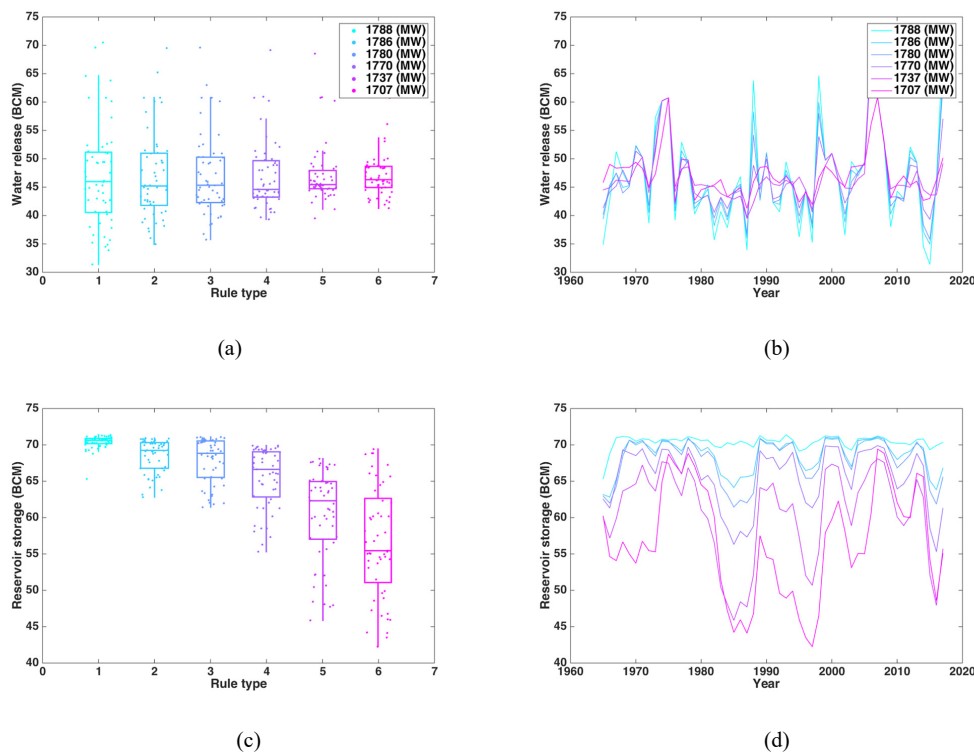

Fig. 6 Boxplots and values of annual reservoir (a)(b) water release and (c)(d) storage for various reservoir operating
rules.



Comparing across rule types, rules with high mean output generate more hydropower mainly in wet years
(Fig. 7). In particular, rule type 1 (1788 MW) can generate approximately 670, 760, and 670 MW more
than rule 6 (1707 MW) in years 1988, 1998, and 2017, respectively. In these years, the annual reservoir
inflow is greater than 65 BCM (Fig. 7). It is worth noting that the annual reservoir inflow in the previous
one or two years (i.e., year 1987, 1997, and 2015) is less than 38 BCM (Fig. 7) and the corresponding
reservoir storage is much higher than in rule types (Fig. 6 (d)). Thus, it can be inferred that rule types with
larger power generation can increase the mean output by releasing less water during dry years to maintain
relatively higher reservoir water levels. In this way, more water will be available and higher head ( $H_t^P$
in equation (1)) can be achieved for future wet years, leading to much more power generation.

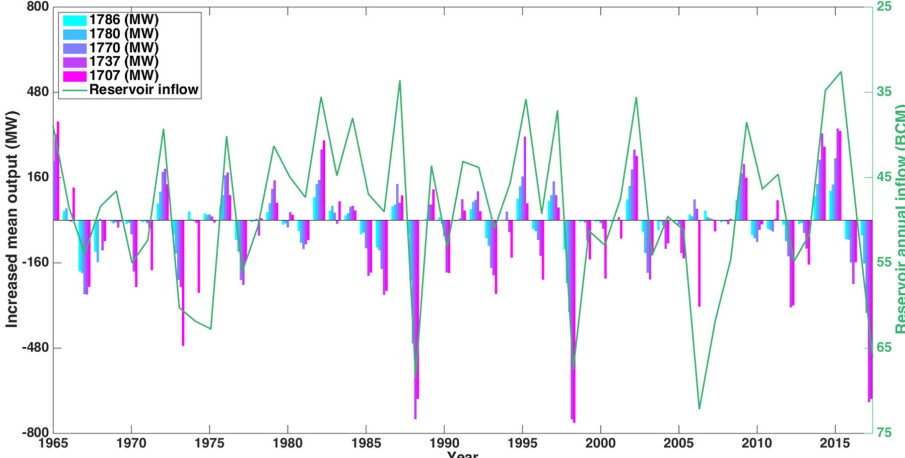


Fig. 7 Net power generation output of various reservoir operating rules compared with rules producing a mean
output of 1788 MW.

However, releasing less water in dry years is not a strategy preferred by downstream countries, which
could further aggravate drought conditions. According to the relationship between annual reservoir inflow



and water release simulated from rule type 1, water release is less than reservoir inflow in most cases (Fig.
8). In comparison, rule type 3 (1780 MW) releases more water than reservoir inflow in dry years. As
power generation decreases further the number of years with reservoir water releases exceeding inflow
increases. Applying a linear regression between annual reservoir inflow and water release (see the lines
in Fig. 8), a drought mitigation policy (equation (10)) can be extracted to constraint annual water release
in reservoir operation. Rule types favoring more power generation generally produce a steeper gradient in
the drought mitigation policy.

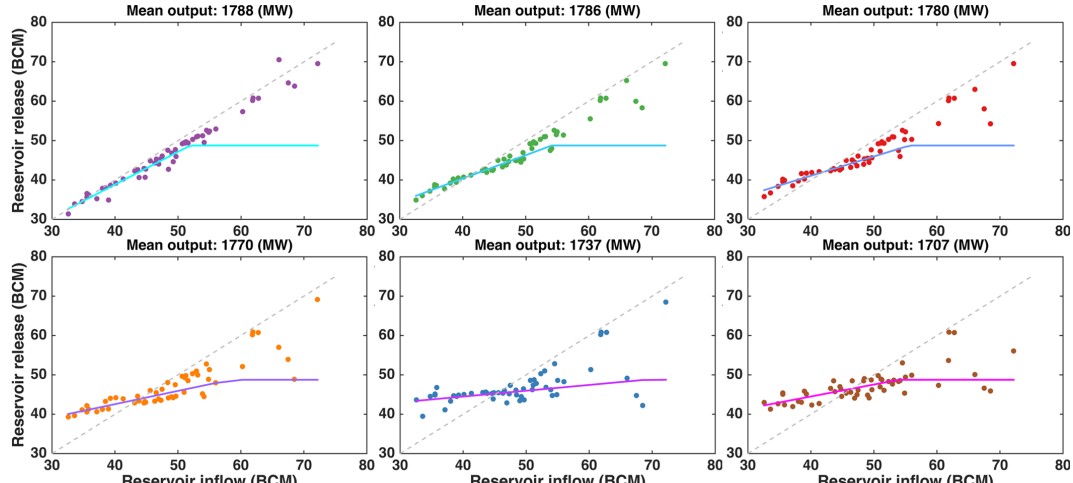


Fig. 8 Relationship between annual reservoir inflow and water release (points) and the corresponding drought
mitigation policy (lines) for various power generation levels.
4.2. **Drought policy selection and analysis**
To select the most suitable drought mitigation policy, both the corresponding power generation and
distribution of annual reservoir water releases need to be evaluated. In general, the distributions of annual
reservoir inflow and releases are significantly different when the reservoir operation is tailored to drought


mitigation. This difference is more pronounced for lower power generation levels (Fig. 9). Considering
low flows, the 10[th] percentile of water releases increases as hydropower generation decreases, from 35.6
BCM for rule type 1 (1788 MW) to 42.7 BCM for rule type 6 (1707 MW).  In contrast, the 10[th] percentile
of annual reservoir inflow is constant at 35.8 BCM. Thus, except for rule type 1 with equal inflow and
outflow volumes, all other rule types ensure that the 10[th] percentile of releases is greater than inflows.
This equates to supplementing downstream flows to address drought conditions when the 10% exceedance
value of annual reservoir inflow is used as the drought threshold. Further, even when reservoir inflow is
less than its 20[th] percentile value, water releases are greater than annual reservoir inflow except for rule
type 1 (Fig. 10). However, when the threshold exceeds the 25[th] percentile, only solutions based on rule
type 3-6 contain annual releases surpassing inflow. These distributions (Fig. 9 and Fig. 10) can provide
critical insights during riparian negotiations regarding trade-offs between power generation and
supplementing downstream flows during drought conditions. Although only six candidate solutions are
illustrated here, more representative solutions may be analyzed in practice.

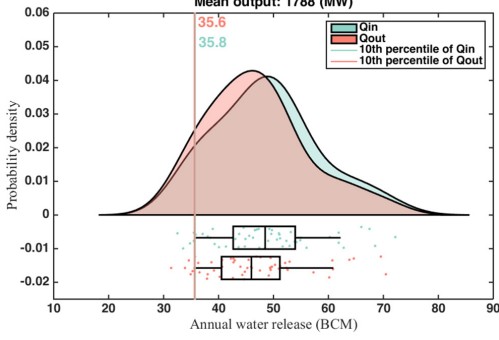
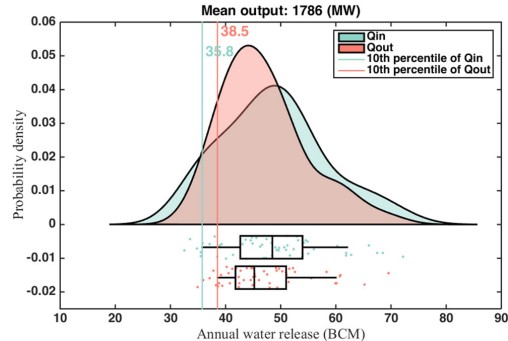



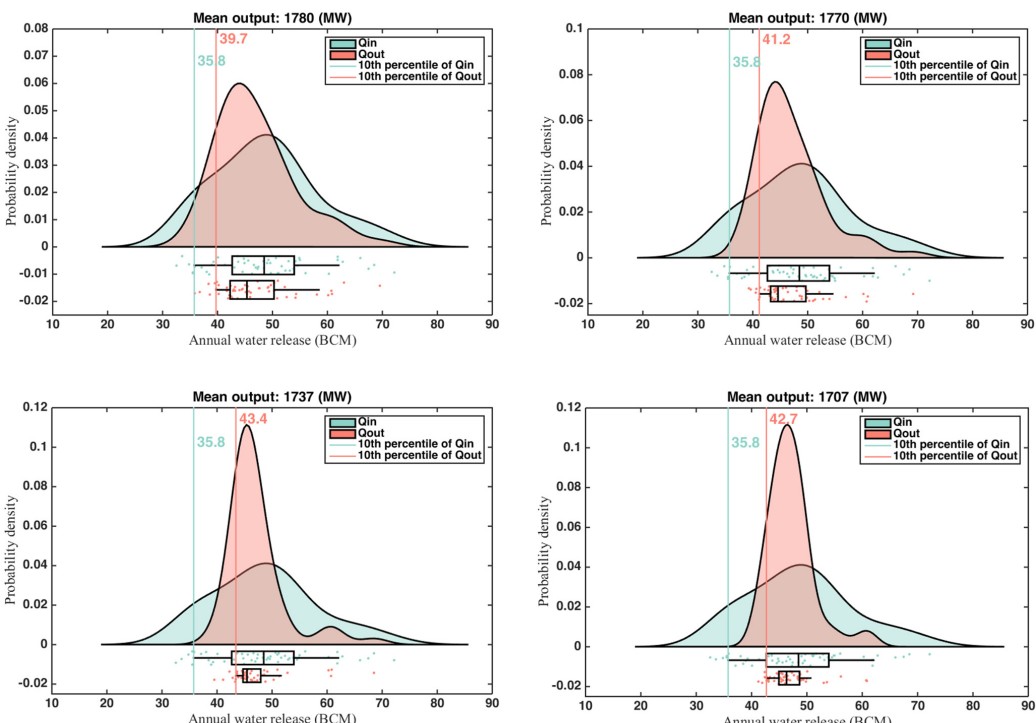

Fig. 9 Kernel distribution of annual reservoir inflow (Qin) and water release (Qout) under different power generation
levels (1965-2017). Vertical lines represent the 10% exceedance value.

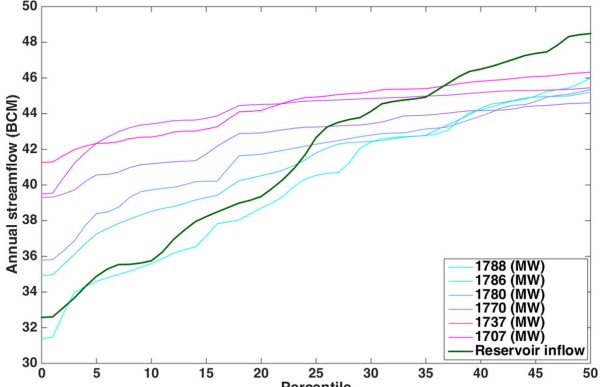


Fig. 10 Percentiles of annual reservoir inflow and water release under various power generation levels.
Incorporating these drought policies (Fig. 8), reservoir operating rules are optimized again for maximum
power generation and minimum deviation of annual release volumes, illustrating varying trade-offs for
drought policies 1-6 (Fig. 11). Drought policies produce similar but not exact hydropower generation as
the original operating rules (e.g. policy 1 original = 1788MW, drought = 1791MW); the standard deviation
of annual releases also does not change significantly. Comparing drought policies producing a high level
of hydropower production (e.g. moving from policy 1 to 2), a small trade-off in production (~4MW) leads
to approximately a 2 BCM decrease in the standard deviation of annual releases.  For lesser hydropower
production policies (e.g. moving from policy 5 to 6), a larger difference of 37MW leads to a smaller (~1
BCM) change in the standard deviation of annual releases.
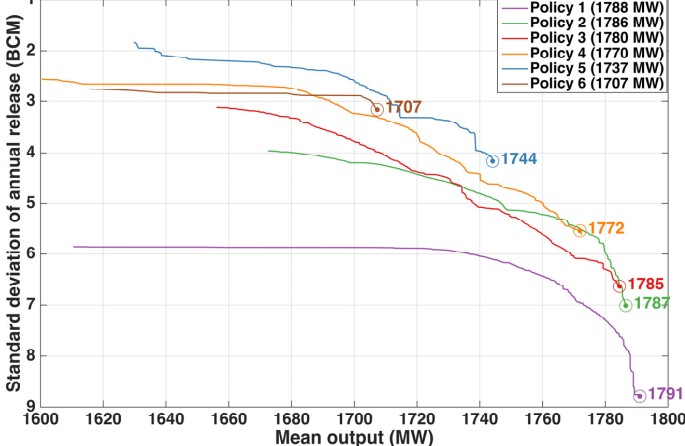

Fig. 11 Multi-objective optimization of reservoir operating rules with drought mitigation policies.
To further evaluate the effectiveness of these drought policies, the mean power output, 10[th] percentile of
annual water releases, and standard deviation of annual water releases obtained from each policy
(corresponding to the Pareto fronts in Fig. 11) are analyzed (Fig. 12).  In Fig. 12, each line presents a set
of reservoir operation results corresponding to a solution from each of the six Pareto fronts in Fig. 11,
based on an expectation of (long-term) monthly average inflow conditions (i.e. not perfect information.)



Hydrology and
Earth System
Sciences



Discussions

338 Predominantly, rules including the drought mitigation policy have higher 10th percentile of annual water

339 release values than original rules. For policies 1 and 2, in addition to notably more 10th percentile annual

340 release values, mean power output does not appreciably drop, thus simultaneously supplementing

341 downstream flows during drought without significant power output losses. Although policies 3-6 provide

342 more 10th percentile of annual releases than the original rules, power output drops; however the standard

343 deviation of annual water releases is significantly less, which indicates that all policies (derived from all

344 power generation levels) can effectively address downstream drought.

345 Since these drought policies are effective even using climatological forecast information, it can be inferred

346 that the effectiveness of drought mitigation policies for the GERD case does not rely on accurate forecast

347 information. This bodes well for other cases lacking accurate hydrological forecasts.

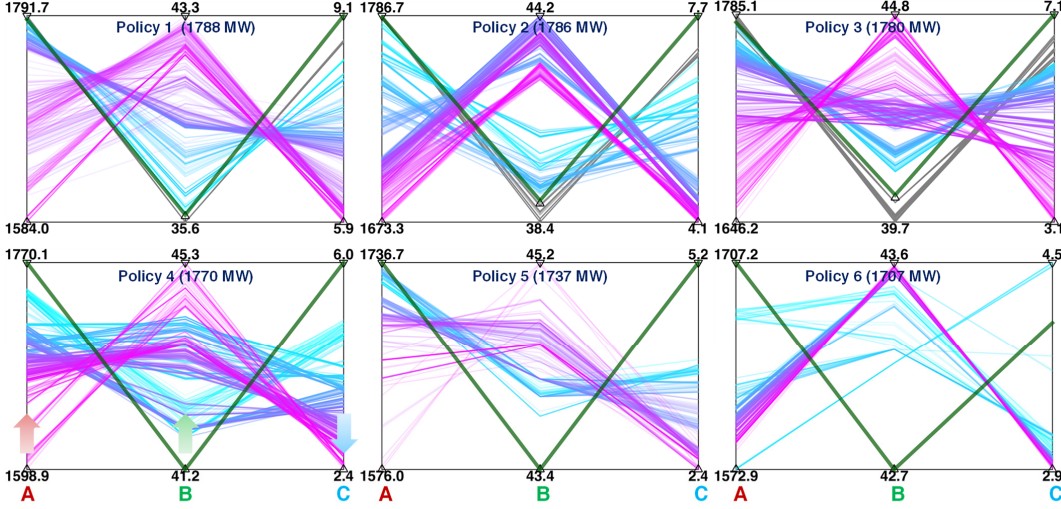


349 Fig. 12 Parallel plots of multiple objectives (A: mean output (MW), B: 10th percentile of annual water release (BCM), C:

350 standard deviation of annual water release (BCM)). The bold green line refers to the reservoir operation without the

351 drought policy.

After re-optimization with the drought policy information included, greater power generation and smaller
values of the standard deviation of annual water releases are produced. More importantly, the re-optimized
rules can fully ensure minimum annual releases under different reservoir inflow levels (Fig. 13).

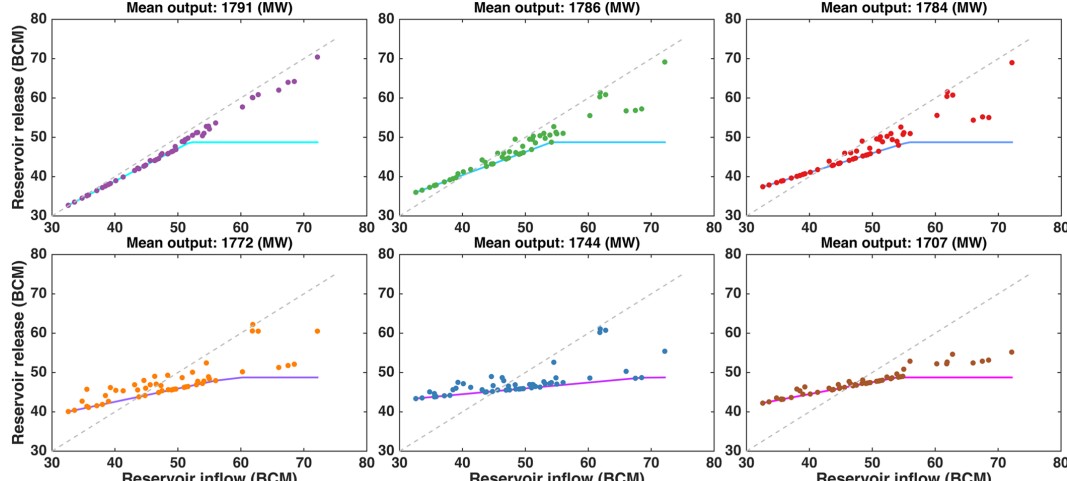


Fig. 13 Relationship between annual reservoir inflow and releases using re-optimized reservoir operating rules;
drought policies represented by lines.
These rules produce slightly more power than the original rules for equivalent standard deviation of annual
release values (Fig. 14(a)) even though they are re-optimized constrained on annual water releases for
drought conditions). Performance of the re-optimized rules, however, mainly depends on the exceedance
parameter $z$ in equation (10); more conservative drought mitigation policies (with larger $z$ values) can
generate more power. Because the trade-off between power generation and the standard deviation of
annual releases is similar between the original rules and drought policy rules ( Fig. 14(b)), it is feasible to
base negotiations on the original rules in this case, as the expected drought policy outcomes are superior.



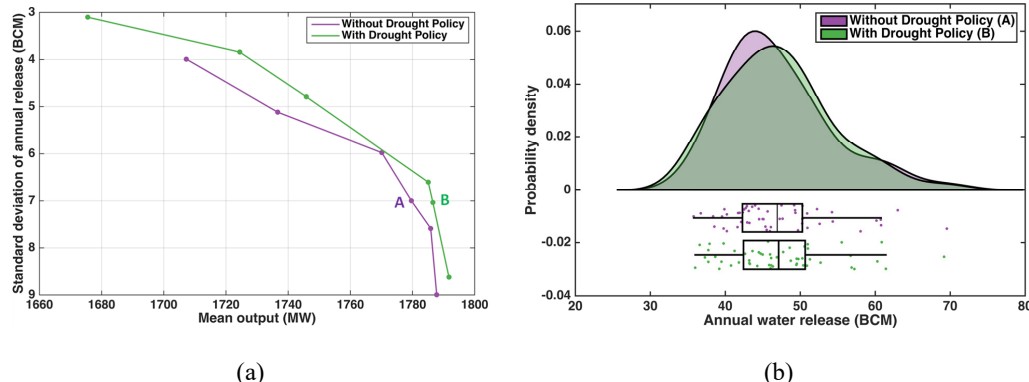

(a)                                        (b)

Fig. 14 (a) Pareto fronts of reservoir operation with and without drought policy; (b) boxplot of reservoir storage for

solution A and B in the Pareto fronts and kernel distribution of annual reservoir water releases with and without the

drought policies.

## 5. Conclusions

Reservoir operations in transboundary river basins are often complex given diverse and potentially

conflicting objectives between upstream and downstream countries. Applying the water-sharing policy

framework proposed here for the Grand Ethiopian Renaissance Dam on the Blue Nile River, we establish

a relationship between downstream and upstream water availability, derive water-sharing policies from

multi-objective optimization results of reservoir operating rules, and analyze the effectiveness of these

policies during drought periods. We demonstrate that a framework incorporating RBF-based rules and a

drought-focused water sharing policy can lead to robust reservoir decision-making. There is a clear trade-

off between power generation and the standard deviation of reservoir releases; however, effective policies

are available to balance this trade-off, even considering drought periods.

This framework here is based on annual flows, however seasonal and monthly scale operations could be

of primary importance in smaller basins or for smaller-capacity reservoirs. Also, many other objectives



and constraints including firm power output, agricultural water supply reliability, and ecosystem functions
could be considered. Future research could explore drought-focused water sharing policies guiding
reservoir operations across multiple time scales simultaneously and the application of seasonal-to-sub-
seasonal inflow forecasts.
**Code and Data Availability Statement**
Some or all data, models, or code that support the findings of this study are available from the
corresponding author upon reasonable request.
**Author contribution**
Guang Yang developed the model code and performed the simulations, visualizations, and original
draft preparation. Paul Block conceptualized the idea and performed data curation and writing- reviewing
and editing.
**Competing interests**
The authors declare that they have no conflicts of interest in this work.
**Acknowledgements**
This work was partially supported by NSF INFEWS award 1639214.

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
