# Peer review of "Water sharing policies conditioned on hydrologic variability to inform reservoir"

_Hydrology and Earth System Sciences, 2021_

## Referee Comment (RC1)

**Title:** Transboundary water sharing policies conditioned on hydrologic variability to inform reservoir operations

**Author(s):** Guang Yang and Paul Block

**MS No.:** hess-2021-72          **MS type:** Research article

**Reviewer:** Michael J Tumbare

1. **General Comments**
   The Article is well written, and is technically and scientifically sound.
2. **Review Comments/Discussion Points**
   a. The scenario relating to the reservoir operation rule curve regards spillage, in which case they could be adverse flooding both upstream and downstream of GERD, has not been dealt with to the same detail and effort as the drought scenario.
   b. It would be interesting to see and compare the results of the resultant water sharing policies if they are applied to, and compared with, other reservoir operation rule curves and water sharing policies for existing dams such as Kariba and Akosombo.
   c. The concepts of "benefit sharing" and "water sharing" are being intermixed in the Article and yet result in different end states. The concept of "benefit sharing" should also be followed to its logical conclusion in the Article, such as optimizing the reservoir operating rules incorporating derived benefits from "benefit sharing" policies and thereafter compare with "water sharing" policies.
   d. How do the drought-focused water sharing policy results compare with the natural minimum flow policy criteria for passage of water to downstream users (Equation 10)?
   e. Page 18, Lines 289 to 290. It is proffered that the passage of the natural minimum flow rule/policy should apply in the case that downstream releases should not be less than the natural minimum flows or inflows into the GERD.
   f. Demonstration of reservoir release benefits in drought years should also be shown and compared with those before the GERD impoundments. In any case, the comparison should not only be based on corresponding power generation alone (Page 19, Lines 302 to 303).
   g. Page 24, Lines 362 to 364. The logic here inclines towards the need for negotiations. If GERD was not there, were they going to be "release" negotiations during droughts or the natural minimum flows would have been expected out of Ethiopia for that drought period?
   h. The final optimal operational rule curve and policy for the GERD is not apparently given.
3. **Technical Corrections**
   Remove third tense throughout the Article e.g. Page 1, Line 9 "we".

---

## Author Comment (AC1)

**Reply Letter**

**Title:** Transboundary water sharing policies conditioned on hydrologic variability to inform reservoir operations

**Author(s):** Guang Yang and Paul Block

**MS No.:** hess-2021-72 **MS type:** Research article

**Reviewer: Michael J Tumbare**

**1. General Comments**

The Article is well written, and is technically and scientifically sound.

Reply: Thank you very much for your positive feedback. The paper has been revised according to your comments.

**2. Review Comments/Discussion Points**

a. The scenario relating to the reservoir operation rule curve regards spillage, in which case there could be adverse flooding both upstream and downstream of GERD, has not been dealt with to the same detail and effort as the drought scenario.

Reply: Regarding spilling, releases are set to be lower than the maximum reservoir inflow during the high-flow season to reduce or eliminate downstream floods (see equation 6 in the original manuscript).

According to the GERD operation results, the annual reservoir water release in wet years is lower than the reservoir inflow (all the points in Figure 13 in the original manuscript are below the 1:1 dashed line when the annual reservoir inflow is greater than the multi-year average: 49 BCM). Our overall drought policy recommendation is for a mean output of 1784 MW (top-right of Figure 13 in the original manuscript), for which the GERD operation has effectively eliminated the extreme low and high flows, and thus benefits flood control. Furthermore, another objective in the optimization of the reservoir operating rules is minimum downstream release deviation, which can help to avoid extremely high and low releases.

To clarify this point, the manuscript has been amended as below.

*To minimize adverse downstream flood conditions due to excess spilling, water release volumes are constrained to be less than the maximum reservoir inflow during the flood season.*

b. It would be interesting to see and compare the results of the resultant water sharing policies if they are applied to, and compared with, other reservoir operation rule curves and water sharing policies for existing dams such as Kariba and Akosombo.

Reply: Thank you for your suggestions. We have compared the results of the proposed (flexible, hydrologic-variability-based) water sharing policies with other conventional water sharing policies in the revised manuscript. The conventional water sharing policy here refers to a "guaranteed quantity at a point," or "minimum flow," strategy. In our case, this would refer to the upstream GERD guaranteeing a fixed amount of water every year or time period. Compacts adopting this strategy in whole or in part include the Colorado River Compact, Arkansas River Basin Compact, and Sabine River Compact, 68 Stat. 690 (1953) (Draper 2006; McCormick 1994).

We have included a comparison and corresponding discussions in the revised manuscript as below.

*The reservoir operation results of the proposed drought policy are compared with those of conventional drought/water sharing policies. A conventional water sharing policy here refers to a "guaranteed quantity" or "minimum flow" strategy, i.e., GERD will guarantee a fixed volume of water release each year. Compacts adopting this strategy in whole or in part include the Colorado River Compact, Arkansas River Basin Compact, and Sabine River Compact, 68 Stat. 690 (1953) (Draper 2006; McCormick 1994). A comparison (Fig. 14) indicates that the flexible drought policy proposed here can generate more power than a conventional (static) drought policy with a similar statistical distribution of water releases. In addition, flexible policies can better mitigate drought conditions (see the kernel distribution as well as $10^{th}$ percentile of water releases in Fig. 14) than static policies for similar power output levels.*

[Figure]

*Fig. 14 Comparison of reservoir operations using flexible and static drought policies based on power generation output and water release distribution analysis. Policy type 1 refers to the comparison with a similar statistical distribution of water releases; Policy type 2-6 refer to comparisons with similar power generation outputs.*

c. The concepts of "benefit sharing" and "water sharing" are being intermixed in the Article and yet result in different end states. The concept of "benefit sharing" should also be followed to its logical conclusion in the Article, such as optimizing the reservoir operating rules incorporating derived benefits from "benefit sharing" policies and thereafter compare with "water sharing" policies.

Reply: Thank you for your comments. Yes, we agree that "benefit sharing" and "water sharing" are unique concepts and should not be intermixed in the Article. Our intention is to mainly focus on "water sharing" instead of "benefit sharing". We mention the "benefits" in the Introduction (Section 1), Models and Methods (Section 3), and Results and Discussion (Section 4.1), but again the intent is to be separate from "water sharing."

In the Introduction, we reference literature regarding "benefit sharing" in transboundary river basins. For example, Arjoon et al. (2016) proposed a benefit-sharing method based on the optimization results from a hydro-economic model and evaluated the value of cooperative water management in the Eastern Nile River basin, in which "benefit sharing" represents developing a sharing strategy for the allocation of monetary benefits from, to, and beyond the river, to ensure basin-wide coordination. In comparison, the "benefit" in other sections refers to the preferences of upstream and downstream in reservoir water release patterns (the upstream prefers releasing less and more water in dry and wet years, respectively for maximum power generation, while the downstream may prefer more evenly distributed releases each year for drought mitigation).

Most benefit sharing policies heavily rely on hydro-economic modeling and cost-benefit analysis (Jeuland et al. 2014), which strives for maximizing the overall aggregated benefit and then allocating the benefit in an equitable way. However, (1) the aggregation of benefits can hide important trade-offs and may increase the risk of droughts for maximum economic benefit; (2) currently, there is no standard that regulates how benefits from various sectors are quantified in Blue Nile basin and it is difficult to develop a mutually agreed-on benefit allocation strategy because of the diversity of stakeholders and their preferences. Also, currently, there is no basin-wide authority in the Blue Nile Basin to enforce benefit allocations (e.g. payments from one country to another), which is another limitation of benefit sharing policies (Arjoon et al. 2016; Dombrowsky 2009). Therefore, we focus on a "water sharing" policy instead of a "benefit sharing" policy in this study. We optimize both power generation and temporal distribution of water releases to account for the trade-off between economic benefits and downstream drought risk, and in this way, drought mitigation becomes an inherent part of the water sharing framework. To avoid confusion of "benefit sharing" and "water sharing" in this study, we have further clarified in the Introduction of the revised manuscript as below.

*Additionally, benefit sharing policies rely heavily on hydro-economic modeling and cost-benefit analysis (Jeuland et al., 2014), which strives to maximize overall aggregated benefits and subsequently allocate benefits in an equitable way. However, (1) the aggregation of benefits can hide important trade-offs and may increase the risk of floods and droughts for maximum economic benefit; (2) there is no standard that regulates how benefits of water use from various sectors (e.g., drinking, agriculture, industry, recreation, and navigation) are quantified and what mechanism should be applied to equitably allocate/share the benefits (Acharya et al., 2020); and (3) there is presently no basin-wide authority to enforce benefit allocations (e.g. payments from one country to another) although institutions such as the Nile Basin Initiative could serve in this role (Arjoon et*

*al., 2016). Thus, water sharing policies considering the trade-off between economic benefits and drought risk, rather than benefit sharing policies based on cooperative operation strategies analysis, are investigated in this study.*

Arjoon, D., Tilmant, A., and Herrmann, M. (2016). "Sharing water and benefits in transboundary river basins." *Hydrology & Earth System Sciences*, 20(6).

Dombrowsky, I. (2009). "Revisiting the potential for benefit sharing in the management of transboundary rivers." *Water Policy*, 11(2), 125-140.

Draper, S. E. "Sharing water in times of scarcity: Guidelines and procedures in the development of effective agreements to share water across political boundaries."

Jeuland, M., Baker, J., Bartlett, R., and Lacombe, G. (2014). "The costs of uncoordinated infrastructure management in multi-reservoir river basins." *Environmental Research Letters*, 9(10), 105006.

McCormick, Z. L. (1994). "INTERSTATE WATER ALLOCATION COMPACTS IN THE WESTERN UNITED STATES – SOME SUGGESTIONS 1." *JAWRA Journal of the American Water Resources Association*, 30(3), 385-395.

d. How do the drought-focused water sharing policy results compare with the natural minimum flow policy criteria for passage of water to downstream users (Equation 10)?

Reply: We have included the comparison between the proposed drought-focused water sharing policy and natural minimum flow policy (static policy) in the revised manuscript (see the comparison in Policy type 1 in Figure 14 as below). It can be seen that the proposed flexible drought policy can produce more power than a static drought policy with a similar statistical distribution of water releases. This is because the flexible policy is derived from optimal reservoir operation results, which tends to produce more power generation. In contrast, the strategy of "guaranteeing a certain flow volume" (which is presented as a horizontal line instead of sloped lines in Figure 13) transfers the risk of water shortages (or hydrologic variability) to the upstream GERD, which will limit GERD's ability to produce more power. Thus, to ensure the same/similar amount of power generation, the 'fixed flow quantity' (release criteria) needs to be lower than the minimum release from the flexible policies (see the lowest scatters of Policy types 2-5 in Figure 14). It can thus be inferred that the static policy will produce less power than the flexible policy when the criteria of the 'fixed flow amount' is the same as the minimum release constraint of flexible policies (Policy type 1 in Figure 14). We have included the analysis in the revised manuscript as below.

*A comparison (Fig. 11) indicates that the flexible drought policy proposed here can generate more power than a conventional (static) drought policy with a similar statistical distribution of water releases. In addition, flexible policies can better mitigate drought conditions (see the kernel distribution as well as $10^{th}$ percentile of water releases in Fig. 11) than static policies for similar*

*power output levels. This is because the flexible policy is derived from optimal reservoir operation results, which tends to generate more power. In contrast, the static policy (which is presented as a horizontal line instead of sloped lines in Fig. 10) transfers the risk of water shortages (or hydrologic variability) completely to the upstream GERD, which will limit GERD's ability to produce more power.*

[Figure]

*Fig. 14 Comparison of reservoir operations using flexible and static drought policies based on power generation output and water release distribution analysis. Policy type 1 refers to the comparison with a similar statistical distribution of water releases; Policy type 2-6 refer to comparisons with similar power generation outputs.*

e. Page 18, Lines 289 to 290. It is proffered that the passage of the natural minimum flow rule/policy should apply in the case that downstream releases should not be less than the natural minimum flows or inflows into the GERD.

Reply: Yes, downstream releases should not be less than the minimum GERD reservoir inflow. This constraint has been included in the revised manuscript and the results have been updated. For example, according to the updated Figure 6 (b) as below, reservoir water releases are equal to or greater than the minimum reservoir inflow (shown with a dashed line).

[Figure]

*Fig. 6 Boxplots and values of … (b) water releases …. dashed line in (b) refers to reservoir inflow. Also, we have updated the corresponding description as below.*

*However, releasing less water in dry years is not a strategy preferred by downstream countries. Although downstream releases are always greater than the minimum natural GERD inflow (which occurs in 2015), releases may clearly be less than natural flow in some other dry years (e.g., 1965 & 1997, see Fig. 6 (b)), which may aggravate drought conditions. According to the relationship between annual reservoir inflow and water release simulated from rule type 1, water release is less than reservoir inflow in most cases (Fig. 7 (a)).*

[Figure]

*Fig. 7 Relationship between annual reservoir inflow and water release (points) and the corresponding drought mitigation policy (lines) for various power generation levels.*

f. Demonstration of reservoir release benefits in drought years should also be shown and compared with those before the GERD impoundments. In any case, the comparison should not only be based on corresponding power generation alone (Page 19, Lines 302 to 303).

Reply: We agree with this suggestion. Reservoir release benefits in drought years have been demonstrated via comparison between the distribution of reservoir releases and inflows (or natural flows) such as in Fig. 9 of the original manuscript. We did not estimate the monetary benefits from reservoir releases by using hydrological-economic simulations as in previous studies (Arjoon et al. 2016). Instead, we use the annual reservoir release amount and the deviation of annual release as a proxy of downstream benefits. More specifically, we compare the statistical distribution of annual reservoir releases and inflows (natural flows). Except for the rule type with a power output of 1788 MW, for all other rule types the 10th percentile of releases is greater than the 10th percentile of annual reservoir inflow (35.8 BCM) (see Figure 9), demonstrating benefits downstream. The comparison of the statistical distribution of annual reservoir inflow (Qin) and release (Qout) from rule type 2 with a mean output of 1786 MW is shown in Fig. 9 and also below, in which vertical lines represent the 10% exceedance value. If this 10% exceedance value of annual reservoir inflow is used as a drought threshold, this rule can be considered effective for drought mitigation (e.g. the 10th percentile of flows increases from 35.8 to 38.5 BCM due to GERD operation). However, this rule will fail to mitigate drought at a threshold of 25% exceedance value (e.g. 25th percentile of releases is less than natural inflow).

[Figure]

Fig. 9 (upper-left) Kernel distribution of annual reservoir inflow (Qin) and water release (Qout) under different power generation levels (1965-2017). Vertical lines represent the 10% exceedance value.

We have further explained this reservoir release benefit in drought years in the revised manuscript as below.

*To select the most suitable drought mitigation policy, both the corresponding power generation and reservoir release benefits in drought years may be evaluated. In this study, annual reservoir release amount and the deviation of annual releases are used as proxies for downstream benefits. For example, if annual releases during drought years is greater than annual reservoir inflow (or*

*natural flow), downstream droughts are partially mitigated.* In general, the *statistical* distributions *of annual reservoir inflow and releases are significantly different when reservoir operations are tailored to drought mitigation. This difference is more pronounced for lower power generation levels (Fig. 9). Considering low flows, the $10^{th}$ percentile of water releases increases as hydropower generation decreases, from 35.6 BCM for rule type 1 (1788 MW) to 42.7 BCM for rule type 6 (1707 MW). Except for rule type 1, all rule types ensure that the $10^{th}$ percentile of releases is greater than* the $10^{th}$ *percentile of annual reservoir inflow (35.8 BCM).*

Arjoon, D., Tilmant, A., and Herrmann, M. (2016). "Sharing water and benefits in transboundary river basins." *Hydrology & Earth System Sciences*, 20(6).

g. Page 24, Lines 362 to 364. The logic here inclines towards the need for negotiations. If GERD was not there, were they going to be "release" negotiations during droughts or the natural minimum flows would have been expected out of Ethiopia for that drought period?

Reply: It is paramount to develop a water sharing policy for the realization of mutual benefits in the management of trans-boundary rivers. If GERD did not exist, "release" negotiations would clearly not be necessary as Ethiopia has no other major on-stream storage capacity. With GERD, of course, negotiations are vitally important, and the proposed drought policy provides intuitive guidance (including the trade-off between power generation and water releases in drought conditions as well as the framework to develop flexible water sharing policies) and maps out what levels of power generation and statistical distributions of releases in drought and non-drought conditions are possible.

Also, we find that the proposed drought policy outperforms the "minimum flow" strategy in drought mitigation and power generation. The limitation of the "minimum flow" strategy is that the risk of water shortage mainly falls to the upstream country and uncertainty of flows increases the risks of the upstream party's ability to meet their obligations.

h. The final optimal operational rule curve and policy for the GERD is not apparently given.

Reply: This study mainly focuses on a framework for deriving operational reservoir water-sharing policies for drought mitigation in transboundary river basins and not in prescribing the optimal outcome. The final optimal reservoir operational policy depends on the negotiation between upstream and downstream countries, at least on how much power generation is expected. For example, according to Figure 13 in the original manuscript (Figure 10 in the revised manuscript as below), if Ethiopia expects to produce a mean power output greater than 1780 MW and downstream countries accept the corresponding distribution of water releases (Figure 9 in the original manuscript), then "Policy 3" (Figure 10c, as below) may be the preferred policy.

[Figure]

*Fig. 10 Relationship between annual reservoir inflow and releases using re-optimized reservoir operating rules; drought policies represented by lines; gray points refer to the inflow and release relationship from which drought policies are derived.*

**3. Technical Corrections**

Remove third tense throughout the Article e.g. Page 1, Line 9 "we".

Reply: Thank you for the suggestion. Third tense and the first-person plural "we" have been corrected with the proper tense in the revised manuscript as below.

*A water-sharing policy framework that incorporates reservoir operating rules optimization based on conflicting uses and natural hydrologic variability, specifically tailored to drought conditions, is proposed. First, the trade-off between downstream and upstream water availability utilizing multi-objective optimization of reservoir operating rules is established. Next, reservoir operation with the candidate (optimal) rules is simulated, followed by their performance evaluations, and the rules selections for balancing water uses. Subsequently, a relationship between the reservoir operations simulated from the selected rules and drought-specific conditions is built to derive water-sharing policies. Finally, the reservoir operating rules are re-optimized to evaluate the effectiveness of the drought-specific water sharing policies. With a case study of the Grand Ethiopian Renaissance Dam (GERD) on the Blue Nile River, it is demonstrated that the derived water sharing policy can balance GERD power generation and downstream releases…*

*In this study, a systemic framework is proposed to derive operational reservoir water-sharing policies using multi-objective optimization for water use conflict mitigation.*

*The Grand Ethiopian Renaissance Dam (GERD) in Ethiopia is selected to demonstrate the framework and illustrate how operational water-sharing strategies,…*

*In this study, GERD reservoir operation rules are developed considering power generation and downstream water release simultaneously to mitigate upstream-downstream water use conflicts,…*

*With* the water-sharing policy framework proposed here for the Grand Ethiopian Renaissance Dam on the Blue Nile River, a relationship between downstream and upstream water availability is established, water-sharing policies *are derived* from multi-objective optimization results of reservoir operating rules, and the effectiveness of these policies during drought periods is analyzed.

*It is demonstrated* that a framework incorporating RBF-based rules and a drought-focused water sharing policy can lead to robust reservoir decision-making.

---

## Author Comment (AC2)

**Reply Letter**

**Title:** Transboundary water sharing policies conditioned on hydrologic variability to inform reservoir operations

**Author(s):** Guang Yang and Paul Block

**MS No.:** hess-2021-72 **MS type:** Research article

Review of HESS-2021-72:

This paper develops optimal water sharing policies for transboundary systems, focusing on the Grand Ethiopian Renaissance Dam. The technical approach is sound, and this system is an important one to study. The results are convincing because they show the tradeoff between minimizing storage variability (i.e. maximizing hydropower) and minimizing release variability for downstream water supply.

I believe that some moderate revisions to the structure and framing would make this paper a stronger contribution to HESS.

Reply: We thank the reviewer for his/her positive comments. The paper has been revised according to your suggestions as described below.

1. My biggest concern is confusion over the methodology, both what is being done, and the reason for the steps. The first step makes sense, optimizing a policy structured as a radial basis function to allow flexible use of information. In my understanding, the steps are: optimize a RBF policy, use it to infer a linear drought constraint, and then use this constraint to re-optimize another RBF policy. Is that correct?

If so, why are these last steps needed? For example, the flow of logic in Figure 3 is very unclear. It seems to repeat optimization steps in places. How is the drought mitigation policy different from the RBF policy, and why is it needed? The RBF is already a function of storage, inflow, and the month. The drought mitigation policy sets minimum annual constraints for water releases. But it seems that this could have been part of the original RBF optimization, and/or enforced in the simulation model to ensure that the constraint is met at all times.

More explanation and structure on these points throughout Sections 1-3 would be very beneficial. While the results are convincing overall, I was not sure that the drought mitigation policy is needed in the end (Fig 14), as there is only a small advantage to these extra steps.

Reply: We thank the Reviewer for highlighting this point, and agree that more clarity will be beneficial. The Reviewer's assessment of the steps involved (first paragraph above) is correct. However, it needs to be noted that the exceedance parameter z (which describes the degree of conservativeness in the policy) in equation 10 is required to obtain the final

linear drought constraint. This is why the second optimization is performed. Initially, for the drought mitigation policy with a mean output of 1770 MW (Figure 4 in the original manuscript), only the linear drought constraint with an exceedance parameter z = 80% can ensure power output greater than 1770 MW (all points in the scatter plot are above the linear constraint line). For this study, the exceedance parameter z in our proposed drought mitigation policy is set as 0%, thus the last step includes re-optimization of RBF policy to verify the performance of the linear drought policy. In the example of Figure 4 (z=0%), it is very possible that the re-optimized maximum mean output could be lower than the original 1770 MW (although it was 1772 MW, according to the 'Policy 4' in Figure 11 of the original manuscript).

Thus, the optimization in the first step is used to find the trade-off between downstream and upstream benefits and build the linear drought policy, and the optimization in the last step is used to validate the drought policy performance for various exceedance levels (equation 10).

To avoid misunderstanding, we have included the reservoir inflow and water release relationship, which is used to infer the drought policy, in the revised manuscript as below (gray scatter plot points), to compare with the re-optimized results.

[Figure]

*Fig. 13 Relationship between annual reservoir inflow and releases using re-optimized reservoir operating rules; drought policies represented by lines; gray points refer to the inflow and release relationship from which drought policies are derived.*

The Reviewer is correct in that the drought mitigation policy sets minimum annual constraints for water releases and it could be part of the original RBF optimization (i.e., we can optimize the RBF policy and the drought mitigation policy simultaneously). However, we believe the results of this one-step optimization are not easy to interpret, considering the complex combinations between RBFs and slopes and intercepts of a linear drought policy. For example, two drought policy lines with similar power generation levels could

have totally different slopes and intercepts and it is then difficult to isolate the impact of the slope or intercept. In contrast, for the development of the drought policies proposed here, the impact of slope on power generation is intuitive (see Figure 8 in the original manuscript).

Indeed, the slope of the drought policy line is highly correlated with variability in reservoir releases (a steeper gradient generally produces more variability in reservoir releases), while the intercept (which is conditional on the exceedance level z) mainly controls the degree of drought mitigation; larger z values indicate higher drought thresholds (see the intersection between the policy line and 1:1 dash line in Fig. 4). In this way we can isolate the impact of these two factors and embed them into upstream-downstream negotiations step-by-step.

We have further clarified why the drought mitigation policy and the re-optimization are advantageous in the revised manuscript as below.

*The drought policy is conditioned on reservoir inflows and releases for a transparent, interpretable, and intuitive process, which is important especially when negotiations are involved. The gradient of the policy line is highly correlated with variability in reservoir releases; in general, as the slope increases, so does the variability in releases. Thus, the parameter $\alpha$ can be estimated from the trade-off between reservoir power generation and downstream water release variability. The exceedance parameter z further controls the degree of drought mitigation; larger z values indicate higher drought thresholds (see the intersection between the policy line and 1:1 dash line in Fig. 4). In the drought policy design, these two parameters can be estimated separately to isolate their impact on drought mitigation performance. This case study mainly focuses on the impact of the first parameter as the exceedance parameter z is eventually set to 0%.*

Additionally, Figure 3 has been updated in the revised manuscript as below.

[Figure]

*Fig. 3 Procedure of drought mitigation policy derivation and evaluation for reservoir operation in transboundary rivers.*

In the revised manuscript, we now also directly optimize the RBF policy conditioned on a conventional water sharing policy (minimum annual water release constraints), with the GERD guaranteeing a fixed amount of water release every year. Compacts adopting this strategy in whole or in part include the Colorado River Compact, Arkansas River Basin Compact, and Sabine River Compact, 68 Stat. 690 (1953) (Draper 2006; McCormick 1994). We have included a comparison and corresponding discussions in the revised manuscript as below.

*The reservoir operation results of the proposed drought policy are compared with those of conventional drought/water sharing policies. A conventional water sharing policy here refers to a "guaranteed quantity" or "minimum flow" strategy, i.e., GERD will guarantee a fixed volume of water release each year. Compacts adopting this strategy in whole or in part include the Colorado River Compact, Arkansas River Basin Compact, and Sabine River Compact, 68 Stat. 690 (1953) (McCormick, 1994;Draper, 2006). A comparison (Fig.*

*11) indicates that the flexible drought policy proposed here can generate more power than a conventional (static) drought policy with a similar statistical distribution of water releases. In addition, flexible policies can better mitigate drought conditions (see the kernel distribution as well as 10th percentile of water releases in Fig. 11) than static policies for similar power output levels.*

[Figure]

*Fig. 11 Comparison of reservoir operations using flexible and static drought policies based on power generation output and water release distribution analysis. Policy type 1 refers to the comparison with a similar statistical distribution of water releases; Policy type 2-6 refer to comparisons with similar power generation outputs.*

2. It is also not clear how the drought mitigation policy uses forecast information, or why the original RBF policy does not. Especially because around Line 345 the results suggest that the forecast information is not very useful in the optimized policies.

Reply: The drought mitigation policy uses annual forecast information by replacing the annual reservoir inflow in the x-axis of figure 4 with the annual reservoir inflow forecast/estimation. The original RBF policy is derived at a monthly scale and thus does not use the annual inflow forecast. The forecast information can be further incorporated into GERD operation by using monthly or annual inflow forecasts as one of the input variables ( $X_t$ in equations 8 and 9) in RBFs-based rules. Considering this study mainly focuses on drought policy, we only use the annual inflow estimation for the evaluation of drought policies.

As illustrated in line 234-238 of the original manuscript, in the last month of each year the annual reservoir inflow estimation $Q_y'^{in}$ will be equal to actual annual inflow $\sum_1^{12} Q_m^{in}$ and the estimated minimum annual release $R_y'^{\min}$ will be $R_y^{\min}$. If $Q_y^{out} < R_y^{\min}$, the reservoir water

release in the last month $Q_{12}^{out}$ will be corrected as $Q_{12}^{out} + \left( R_y^{\min} - Q_y^{out} \right)$ and the $Q_y^{out}$ will be equal to $R_y^{\min}$. Thus annual reservoir release $Q_y^{out}$ will be always greater than or equal to the specified minimum reservoir water release $R_y^{\min}$ and it can be inferred that the minimum annual release $R_y^{\min}$ is mainly determined by the policy parameters $\alpha$, $\beta$, and z, rather than forecast accuracy.

We have further explained in the revised manuscript as below.

*The estimated variables $R_y'^{\min}$, $Q_y'^{out}$, and $\sum\limits_{m}^{12} Q_m'^{in}$ are updated in each time step. In the last month of each year, the annual reservoir inflow estimation $Q_y'^{in}$ will be equal to actual annual inflow $\sum\limits_{1}^{12} Q_m^{in}$ and the estimated minimum annual release $R_y'^{\min}$ will be $R_y^{\min}$. If $Q_y^{out} < R_y^{\min}$, the reservoir water release in the last month $Q_{12}^{out}$ will be corrected as $Q_{12}^{out} + \left( R_y^{\min} - Q_y^{out} \right)$ and the $Q_y^{out}$ will be equal to $R_y^{\min}$. Thus annual reservoir release $Q_y^{out}$ will always be greater than or equal to the specified minimum reservoir water release $R_y^{\min}$ and it can be inferred that the minimum annual release $R_y^{\min}$ is mainly determined by the policy parameters $\alpha$, $\beta$, and z, rather than forecast accuracy. As illustrated in equation (10) and Fig. 4, the minimum annual reservoir release can be estimated from the annual reservoir inflow after the drought policy line is determined. Considering actual annual reservoir inflow will not be available until the last month of each year, the annual reservoir inflow forecast is used instead.*

3. Another concern is the novelty of the approach. The study follows best practices for direct policy search (DPS) and arrives at a convincing result. However, I believe this approach has been used for transboundary systems before. It seems the new component of this study is deriving water-sharing policies (i.e. annual linear constraints) that are specific to drought periods. This may be a novel contribution, but as in point (1) it is not clear why this needs to be done here. The link could also be stronger between the linear constraint and the idea of a negotiated water-sharing policy between transboundary stakeholders.

Last, if the novelty relates to transboundary basins, is there any component of the methodology that is specifically designed for this case? The contribution may be more general, although the transboundary application is critically important.

Reply: Thank you for your suggestions. The novelty of this work is the water-sharing policy design, which combines trade-off analysis of upstream-downstream benefits and drought mitigation based on hydrological variability. Traditional water-sharing policies generally control the reservoir water release or storage in a static way, for example, a "storage limitation" strategy limits the amount of water that an upstream entity may impound annually, seasonally, etc., while a "minimum flow" strategy guarantees a fixed volume of releases every year or other time period. In this way, most of the risk in water shortages or hydrological variability will fall upon the upstream or downstream parties and

the trade-off of upstream-downstream benefit/risk is rarely considered in water sharing policy design.

To support water sharing in a more flexible way, some literature develops benefits sharing policies (Teasley and McKinney 2011) and strategies (Degefu et al. 2016; Li et al. 2019; Wheeler et al. 2016) based on cooperative water resources operations and simulations. These policies and strategies rely heavily on hydro-economic modeling and cost-benefit analysis (Jeuland et al. 2014), which strives to maximize overall aggregated benefits and subsequently allocate benefits in an equitable way. However, (1) the aggregation of benefits can hide important trade-offs and may increase the risk of floods and droughts for maximum economic benefit; (2) there is no standard that regulates how benefits of water use from various sectors (e.g., drinking, agriculture, industry, recreation, and navigation) are quantified and what mechanism should be applied to equitably allocate/share the benefits (Acharya et al. 2020); and (3) there is presently no basin-wide authority to enforce benefit allocations (e.g. payments from one country to another) although institutions such as the Nile Basin Initiative could serve in this role (Arjoon et al. 2016). Thus, water sharing policies considering the trade-off between economic benefits and drought risk, rather than benefit sharing policies based on cooperative operation strategies analysis, are investigated in this study.

The water sharing policy proposed here inherits the flexibility of the aforementioned benefit sharing policies by incorporating reservoir operation optimization and maintains the intuitiveness of traditional water-sharing policies by informing reservoir operations with an interpretable and intuitive linear-regression-based rule.

We have further clarified the novelty of this work in the revised manuscript as below.

[revised manuscript text omitted]

4. The results are a bit too long, with 14 figures. These could be condensed to sharpen the contributions. In my opinion a few figures that might be removed are: Fig 7, Fig 9 (previous figures already show the change in standard deviation), and Fig 12.

Reply: Thank you for your suggestions. In the revised manuscript, we have further sharpened the contributions, removing Fig. 7 and Fig. 12, and moving Fig. 9 to the Supplemental Data.

5. In the introduction and methods, there are several places where the references are grouped together in long lists. It would be stronger to highlight individual contributions from these previous studies where possible.

A minor point about the introduction: clearly simulation-optimization is relevant, but why is policy fitting relevant to this study?

Reply: Thank you for your suggestions. We have shortened or divided these grouped references and highlight individual contributions in the revised manuscript. For example:

*Thus in recent decades, many models and strategies have been investigated to inform and improve reservoir operation decision-making (Chaves and Chang, 2008;Cancelliere et al., 2002;Herman and Giuliani, 2018;Karamouz and Houck, 1982;Giuliani et al., 2014;Oliveira and Loucks, 1997). For example, Karamouz and Houck (1982) optimize monthly reservoir releases by deterministic dynamic programming and build a linear reservoir operation model conditioned on the relationship between optimal releases and reservoir state variables. Cancelliere et al. (2002) build a non-linear reservoir operation model by using neural network techniques to improve reservoir irrigation water supply during drought conditions. Herman and Giuliani (2018) design a tree-based policy which is flexible and interpretable for reservoir operation over multiple timescales.*

Considering simulation-optimization is more relevant than policy fitting to this study, we have removed the highlighted references about policy fitting in the revised manuscript. Policy fitting requires an optimal set of reservoir inflows, storages, and releases and the fitted rules generally perform worse than simulation-optimization-based rules, however, the rules derivation process is interpretable and intuitive when optimal reservoir decision-making is highly correlated with state variables. This is the main reason why policy fitting is used to develop the linear drought policy in this study. We have further explained this reasoning in the revised manuscript as below.

In Introduction:

*A policy fitting approach requires an optimal set of reservoir inflows, storages, and releases and its effectiveness highly depends on the performance of the optimized reservoir operation model; however, the rules derivation is interpretable and intuitive when optimal reservoir decision-making is highly correlated with state variables. In contrast, simulation-optimization-based approaches do not rely on existing optimal reservoir operations and thus it is generally more flexible than fitting-based rules.*

*The drought policy is conditioned on reservoir inflows and releases for a transparent, interpretable, and intuitive process, which is important especially when negotiations are involved.*

6. There are several possible discussion points that could be included briefly. First, the tradeoffs found in this study are based on a period of historical data, but what about the future? Do we expect patterns of variability to be similar? Second, is there a way to better understand the characteristics of the optimized policies so that they could be reported to stakeholders, for example? These can be points for future work but deserve some discussion.

Reply: Thank you for your suggestions. Yes, the trade-off between power generation and water release variability and the relationship between reservoir inflows and releases are based on historical data. Both of them can be affected by land use or climate changes, and if significant, the water sharing policy may need to be adjusted accordingly. Clearly, there still exists a trade-off between reservoir power generation and water release variability under changing conditions, which can still be used to inform drought policy design, and the linear feature of the drought policy makes it relatively easy to adjust. It is very important to connect the characteristics of a water sharing policy with the trade-off between reservoir storage and releases. In this study, greater variability in releases will lead to a steeper gradient of the drought policy line. These types of drought policy characteristics can provide guidance for stakeholders to effectively adjust the water sharing policy.

We have included the following discussion in the revised manuscript as below.

*It is worth noting that the trade-off between power generation and water release variability and the relationship between reservoir inflows and releases are based on historical data and may be affected by future changes in land use, climate, etc. Thus, the water sharing policy may need to be adjusted accordingly to better mitigate drought conditions in the future. However, there will always exist a trade-off between reservoir power generation and water release variability, which can be used to inform drought policy design, and the linear feature of the drought policy makes it relatively easy to adjust. It is very important to connect the characteristics of a water sharing policy with the trade-off between reservoir storage and releases. In this study, greater variability in releases leads to a steeper gradient of the drought policy line. These types of drought policy characteristics can provide guidance for stakeholders to effectively adjust the water sharing policy. Thus, the interpretable drought policy proposed here can enhance the understanding of water sharing and promote multilateral negotiations between upstream and downstream countries.*

---

## Author Comment (AC3)

**Reply Letter**

**Title:** Transboundary water sharing policies conditioned on hydrologic variability to inform reservoir operations

**Author(s):** Guang Yang and Paul Block

**MS No.:** hess-2021-72 **MS type:** Research article

Review of HESS-2021-72:

This paper presents a water-sharing policy framework that incorporates reservoir operating rules optimization based on conflicting uses and hydrologic variability, specifically tailer to drought conditions. The framework is illustrated using GERD as a case study. The results clearly show the trade-off between annual hydropower generation and the inter-annual variability of releases. The paper is well-written and the topic should be of interest to both researchers and practitioners. As explained below, my main concern is with the methodology, which seems to be overly "complex" given the relative simplicity of the case study.

Reply: Thank you very much for your positive comments. We have revised the paper according to your suggestions and provided a point-by-point response to your concerns as below.

Comments/questions:

1. The emphasis is put on average annual power output. However, power companies are also concerned by the firm energy, i.e. the energy output that you can guarantee 90% or 95% of the time. Can you show us what the trade-off would look like when the average energy output is replaced by the firm energy?

Reply: Thank you for your suggestions. We agree with the importance of considering firm energy and due to space limitations did not include it in the original version. Comparing the firm energy output (at a guarantee of 90%) with the standard deviation of releases, we obtain the Pareto front as below.

[Figure]

*Fig. S1 Pareto front for maximum firm output (90% guarantee) and minimum annual water release variance.*

We have added this trade-off in the supplementary data of the revised manuscript. Additionally, we have amended the following text:

*Thus downstream countries may benefit more from reservoir operating rules favoring smaller $Std\left(Q_y^{out}\right)$ in drought conditions; this trade-off between power generation and $Std\left(Q_y^{out}\right)$ can be used to balance GERD power generation and downstream water use benefits. There also exists a trade-off between $Std\left(Q_y^{out}\right)$ and other power indicators such as firm output (see Fig. S1 in Appendix S1).*

It needs to be noted that the firm output of GERD is mainly determined by the operation during January and April (see Figure 2 below). In contrast, both mean output and annual release variance are calculated from the operation results during the full year. Also, releasing more water downstream during dry seasons can - to some degree - increase the firm output. Thus, the trade-off between firm output and annual release variance is not stark. To better illustrate the trade-off between upstream and downstream benefits, we focus on the mean output in this study. Future studies could consider the requirement of firm energy as a constraint in GERD operations.

[Figure]

*Fig. 2 Monthly inflow into the GERD reservoir during 1965-2017.*

2. Does the term "water releases" include turbined outflows AND spillage losses?

Reply: Yes, "water releases" in this study includes both the turbine outflows and excessive spillages. Although spilling water in the flood season is not used for power generation, it can be used downstream. Also, to minimize adverse downstream flood conditions due to excess spilling, water release volumes are constrained to be less than the maximum reservoir inflow during the flood season (see equation 6 in the original manuscript). We have further clarified this in the revised manuscript as below.

Section 2.1:

*In this study, GERD reservoir operation rules are developed considering power generation and downstream water release (including turbine outflows and spillage losses) simultaneously to mitigate upstream-downstream water use conflicts, particularly tailored to drought periods.*

Section 3.1:

*where $Q_y^{out}$ is the reservoir water release (which includes turbine outflows and spillage losses) in year y and $\bar{Q}_y^{out}$ and $Std\left(Q_y^{out}\right)$ are the mean value and standard deviation of reservoir annual water release across all operational years Y, respectively.*

3. When minimizing the variance of water releases, do you end up to a point where the energy output starts decreasing due to excessive spillages losses?

Reply: Thank you for your comments. Considering evaporation loss is much smaller than reservoir inflow, the total volume of water release (during 1965-2017) is approximately the same as the total GERD inflow volume. It can be inferred that minimizing the variance of annual water release will lead to less releases in wet years and thus less excessive spillages losses (which mainly occurs in wet years). We have included the spillage losses of various reservoir operating rules in Figure 6 of the original manuscript as below. It

shows that monthly rules with less variance of water releases (less power output) produce less spillage than other rules. In contrast, spillage loss from rules with a mean output of 1788 MW (rule type 1, with greatest annual release variance) occurs more frequently (and the magnitude is also greater) than other rules.

It needs to be noted that we are optimizing reservoir operating rules not only for minimum annual release variance, but also for maximum monthly mean output. The latter objective will to some degree avoid excessive spillage losses in dry seasons and years. As shown in Figure 11 of the original manuscript (and copied below), the final reservoir operating rules are optimized for maximum mean output, in which the drought policy is used as a constraint, therefore large excessive spillages are unlikely. Thus, when minimizing the variance of water releases, the energy output starts decreasing mainly due to the lower reservoir water level, rather than excessive spillages losses. Also, releases are set to be lower than the maximum reservoir inflow during the high-flow season to reduce or eliminate downstream floods/spilling (see equation 6 in the original manuscript).

[Figure]

[Figure]

(e)

*Fig. 6 Boxplots and values of annual reservoir (a)(b) water releases, (c)(d) storages, and (e) spillages for various reservoir operating rules; dished line in (b) refers to reservoir inflow.*

[Figure]

*Fig. 11 Multi-objective optimization of reservoir operating rules with drought mitigation policies.*

4. My main concern. Why didn't you constrain the operating rule with a minimum "water release" (or minimum deviation from a target release) in the first step of the methodology and construct your Pareto front by varying that minimum like in the traditional constraint method in MOP? The system is small (just one reservoir) and it looks like to me that the Pareto front could be traced out using mathematical programming techniques in a MO framework. In my opinion the introduction must be revised to better explain why that framework was proposed instead of traditional MOP approaches.

Reply: Thank you for your comments. This is an interesting point raised by the reviewer because it's true that the drought mitigation can be considered by constraining the operating rule with a minimum "water release" (or minimum deviation from a target release). This type of "minimum water release" strategy has been applied in whole or in part in the Colorado River Compact, Arkansas River Basin Compact, and Sabine River Compact, 68 Stat. 690 (1953) (Draper 2006; McCormick 1994). We have also optimized the reservoir operating rules with the constraint of minimum GERD "water release" each year and compared this traditional constraint method with our proposed drought policy. We find that the flexible drought

policy proposed here can generate more power than the traditional constraint method with a similar statistical distribution of water releases. We have included the results in the revised manuscript as below.

*The reservoir operation results of the proposed drought policy are compared with those of conventional drought/water sharing policies. A conventional water sharing policy here refers to a "guaranteed quantity" or "minimum flow" strategy, i.e., GERD will guarantee a fixed volume of water release each year. Compacts adopting this strategy in whole or in part include the Colorado River Compact, Arkansas River Basin Compact, and Sabine River Compact, 68 Stat. 690 (1953) (McCormick, 1994;Draper, 2006). A comparison (Fig. 11) indicates that the flexible drought policy proposed here can generate more power than a conventional (static) drought policy with a similar statistical distribution of water releases. In addition, flexible policies can better mitigate drought conditions (see the kernel distribution as well as 10th percentile of water releases in Fig. 11) than static policies for similar power output levels. This is because the flexible policy is derived from optimal reservoir operation results, which tends to generate more power. In contrast, the static policy (which is presented as a horizontal line instead of sloped lines in Fig. 10) transfers the risk of water shortages (or hydrologic variability) completely to the upstream GERD, which will limit GERD's ability to produce more power.*

[Figure]

*Fig. 11 Comparison of reservoir operations using flexible and static drought policies based on power generation output and water release distribution analysis. Policy type 1 refers to the comparison with a similar statistical distribution of water releases; Policy type 2-6 refer to comparisons with similar power generation outputs.*

We agree that the single-reservoir operation problem can be solved by using mathematical programming techniques such as non-linear programming and dynamic programming. However, the main purpose of this study is not to obtain optimal solutions of reservoir operation problems with specific objectives and constraints, instead, we are proposing a framework to derive an intuitive, interpretable, and flexible water sharing policy which can be incorporated into flexible reservoir operating rules. More specifically, we are not minimizing the variance of water releases to select final reservoir operating rules and associated drought policy; instead, we use the release variance as a proxy of downstream benefits to understand the

trade-off between GERD power generation and downstream releases to support negotiations between upstream and downstream stakeholders. After negotiation, the selected (mutually agreed) point in the Pareto front in Figure 5(a) of the original manuscript can be used to infer a linear drought constraint to further derive reservoir operating rules.

To better illustrate the procedure of water sharing policy derivation, we have updated the Figure 3 in the revised manuscript as below.

[Figure]

*Fig. 3 Procedure of drought mitigation policy derivation and evaluation for reservoir operation.*

The procedure of water sharing policy design looks overly "complex" mainly because both the derivation and evaluation of the policy are included (two optimization steps are involved). As shown in the updated Figure 3, optimization in the first step is used to find the trade-off between downstream and upstream benefits and build the linear drought policy, and optimization in the last step is used to validate the drought policy performance for various exceedance levels (equation 10). Also, we consider the potential to inform upstream-downstream negotiations in water sharing policy design through a transparent, flexible, and intuitive process. We have further clarified the necessity of these steps in the revised manuscript as below.

*The drought policy is conditioned on reservoir inflows and releases for a transparent, interpretable, and intuitive process, which is important especially when negotiations are involved. The gradient of the policy line is highly correlated with variability in reservoir releases; in general, as the slope increases, so does the variability in releases. Thus, the parameter $\alpha$ can be estimated from the trade-off between reservoir power generation and downstream water release variability. The exceedance parameter z further controls the degree of drought mitigation; larger z values indicate higher drought thresholds (see the intersection between the policy line and 1:1 dash line in Fig. 4). In the drought policy design, these two parameters can be estimated separately to isolate their impact on drought mitigation performance. This case study mainly focuses on the impact of the first parameter as the exceedance parameter z is eventually set to 0%.*

To better integrate the RBFs-based reservoir operating rules and drought policy, we infer the linear drought constraint by optimizing RBFs-based rules (in which reservoir decision-making is only informed by seasonal information, current reservoir storage and inflow) instead of using a 'perfect' reservoir decision-making series optimized from dynamic programming. In that way, the maximum power generation of a drought policy (see selected points '1791', '1787', and '1785', etc. in Figure 11 of the original manuscript) will be similar to the power generation of the original reservoir operating rules (see Figure 5(a) in the original manuscript). Also, using the simulation-optimization method, instead of mathematical programming techniques to obtain the Pareto front in this case, offers the additional advantage of demonstrating a method that can be applied to reservoir systems with more state variables.

To better clarify the novelty of this work and explain why this water sharing framework was proposed, we have revised the manuscript as below.

*Most previous studies focus on illustrating the importance of a cooperative strategy through water system optimization and simulation (Dombrowsky, 2009;Tilmant and Kinzelbach, 2012) and evaluating the benefits of cooperative operation in transboundary river basins (Goor et al., 2010;Anghileri et al., 2013;Uitto and Duda, 2002;Luchner et al., 2019). There is less literature (Wheeler et al., 2016;Li et al., 2019;Degefu et al., 2016;Teasley and McKinney, 2011)., however, addressing strategies for reaching an agreement or consensus on water resources development amongst downstream and upstream riparian countries in transboundary river basins. Also, although cooperation in transboundary river basins can result in a win-win situation for both downstream and upstream stakeholders, cooperative water use strategies are obstructed by single-sector interests, especially when long-term commitments are involved (Wu and Whittington, 2006). More specifically, it is often difficult to achieve a mutually agreed-on cooperation strategy given divergent solution preferences by stakeholders.*

*Additionally, benefit sharing policies rely heavily on hydro-economic modeling and cost-benefit analysis (Jeuland et al., 2014), which strives to maximize overall aggregated benefits and subsequently allocate benefits in an equitable way. However, (1) the aggregation of benefits can hide important trade-offs and may increase the risk of floods and droughts for maximum economic benefit; (2) there is no standard that regulates how benefits of water use from various sectors (e.g., drinking, agriculture, industry, recreation, and navigation) are quantified and what mechanism should be applied to equitably allocate/share the benefits (Acharya et al., 2020); and (3) there is presently no basin-wide authority to enforce benefit allocations (e.g. payments from one country to another) although institutions such as the Nile Basin Initiative could serve in this role (Arjoon et al., 2016). Thus, water sharing policies considering the trade-off between economic benefits and drought risk, rather than benefit sharing policies based on*

*cooperative operation strategies analysis, are investigated in this study. The policies will be flexible, interpretable, and more importantly drought-focused such that downstream drought mitigation will become an inherent part of the water sharing framework.*

*In this study, a systemic framework is proposed to derive operational reservoir water-sharing policies using multi-objective optimization for water use conflict mitigation. Specifically, (1) optimize reservoir operating rules and establish trade-off between upstream benefits and downstream drought risks, (2) simulate reservoir operation with the candidate (optimal) rules, evaluate performance, and select the most suitable rules for balancing benefits, (3) derive water-sharing policies conditioned on reservoir operations and water availability, and (4) re-optimize reservoir operating rules incorporating derived water-sharing policies to evaluate effectiveness and performance. The drought-focused water-sharing policies are interpretable as they are derived from and evaluated on reservoir operation simulations from existing optimal rules. Further, the policies are considered flexible by offering opportunities for informing upstream-downstream negotiations.*

5. Line 90. Please check the paper from Teasley and McKinney, JWRPM, 2011 on water and benefits sharing in the Aral Sea Basin.

Reply: Thank you for your reminder. The paper from Teasley and McKinney, JWRPM, 2011 develops a draft agreement on the allocation of water and energy resources based on cooperative operation and benefit-sharing. We have included the paper in the introduction of the revised manuscript as below.

*There is less literature (Wheeler et al., 2016;Li et al., 2019;Degefu et al., 2016;Teasley and McKinney, 2011), however, addressing strategies for reaching an agreement or consensus on water resources development amongst downstream and upstream riparian countries in transboundary river basins.*

In this work, we are developing water sharing policies considering the trade-off between economic benefits and drought risk, which is different from a water allocation strategy or agreement based on benefit-sharing as in this JWRPM paper. It needs to be noted that policies and strategies based on cooperative operation and benefit-sharing rely heavily on hydro-economic modeling and cost-benefit analysis (Jeuland et al. 2014), which strives to maximize overall aggregated benefits and subsequently allocate benefits in an equitable way. However, (1) the aggregation of benefits can hide important trade-offs and may increase the risk of floods and droughts for maximum economic benefit; (2) there is no standard that regulates how benefits of water use from various sectors (e.g., drinking, agriculture, industry, recreation, and navigation) are quantified and what mechanism should be applied to equitably allocate/share the benefits (Acharya et al. 2020); and (3) there is presently no basin-wide authority to enforce benefit allocations (e.g. payments from one country to another) although institutions such as the Nile Basin Initiative could serve in this role (Arjoon et al. 2016). Thus, we develop water sharing policies considering the trade-off between economic benefits and drought risk, rather than benefit sharing policies based on cooperative operation strategies analysis in this study.

Acharya, V., Halanaik, B., Ramaprasad, A., Swamy, T. K., Singai, C. B., and Syn, T. (2020). "Transboundary sharing of river water: Informating the policies." *River Research and Applications*, 36(1), 161-170.

Arjoon, D., Tilmant, A., and Herrmann, M. (2016). "Sharing water and benefits in transboundary river basins." *Hydrology & Earth System Sciences*, 20(6).

Draper, S. E. "Sharing water in times of scarcity: Guidelines and procedures in the development of effective agreements to share water across political boundaries."

Jeuland, M., Baker, J., Bartlett, R., and Lacombe, G. (2014). "The costs of uncoordinated infrastructure management in multi-reservoir river basins." *Environmental Research Letters*, 9(10), 105006.

McCormick, Z. L. (1994). "INTERSTATE WATER ALLOCATION COMPACTS IN THE WESTERN UNITED STATES-SOME SUGGESTIONS 1." *JAWRA Journal of the American Water Resources Association*, 30(3), 385-395.

6. I have a small gripe with the title. The methodology is actually applicable to any reservoir and it is not limited to transboundary river basins. Please remove "transboundary" from the title and revise the text accordingly.

Reply: Thank you for your suggestions. We agree to change the title to "Water sharing policies conditioned on hydrologic variability to inform reservoir operations" and revise the text accordingly.

7. Line 56. A wide variety of physiographic conditions is not limited to transboundary river basins!

Reply: Yes, we agree that a wide variety physiographic conditions is not limited to transboundary river basins. Considering this study does not include the impact of various physiographic conditions in reservoir operation, we have removed this in the revised manuscript as below.

*Reservoir operations in transboundary river basins are necessarily more complex given a wide variety of social, political, economic, and cultural, and physiographic conditions (Zeitoun and Mirumachi, 2008).*

8. Figure 12. Could you also include spillages losses and evaporation losses? Keeping the water level as high and as constant as possible will likely increases these two losses, up to a point where they can negatively impact the power output and the total outflows.

Reply: Thank you for your suggestions. We have included spillages losses and evaporation losses in the revised manuscript as below. According to Figure S3, keeping high reservoir water level (greater mean output) will certainly increase both the frequency and the amount of spillage and evaporation losses. Thus, we mitigate this negative affect by optimizing the reservoir operating rules (monthly RBFs-based on direct policy search rules) for maximum power generation (see the selected points in Figure 11 of the original manuscript). Our overall drought policy recommendation is for "Policy 3" in Figure S3, for which spillages losses and evaporation losses are effectively eliminated.

[Figure]

*Fig. S3 Parallel plots of multiple objectives (A: mean output (MW), B: 10th percentile of annual water release (BCM), C: standard deviation of annual water release (BCM), D: spillage loss (BCM), E: evaporation loss (BCM)). The bold green line refers to the reservoir operation without the drought policy.*

---

## Author Comment (AC5)

**Reply Letter**

**Title:** Transboundary water sharing policies conditioned on hydrologic variability to inform reservoir operations

**Author(s):** Guang Yang and Paul Block

**MS No.:** hess-2021-72 **MS type:** Research article

**General comments**

The article is well-written, scientifically sound and within the scope of the journal.

Reply: Thank you very much for your positive feedback. We specifically address your comments below, particularly focusing on providing more details about constraints and experiment settings for GERD reservoir operations.

**Detailed comments**

1. Following are the aspects that could be incorporated into the article.

The actual values of the constraints defined in equation 4 to 7 need to be specified. This is especially pertinent because the range of optimal power generation obtained in the study range from 1707 to 1788 MW while the estimated power generation of 15130GWH per year which implies an average power generation of 4202 MW. The reservoir storage trajectories on Figure 6d reveal an assumed maximum storage of 74 billion cubic metres which is the intended capacity of GERD. The article does not indicate whether the upper power generation limit (PU$_t$in equation 7) equals the intended installation capacity of 5000 MW. On Figure 6d, the reservoir trajectory for a power output of 1788MW is very high and close to full storage for most of the period suggesting that achieving 15130GWH (4202 MW) would require more of what has been simulated as spillage to run through the turbines to generate more electricity. A discussion of how the analysis here relates to the intended installed capacity and power generation would enhance the relevance of the article to the practical transboundary issues regarding the operation of the GERD.

Reply: Thank you for your suggestions. We have specified actual values of the constraints defined in equation 4 to 7 in the revised manuscript as below.

*The reservoir structural and operational constraints can be expressed as:*

$$S^{min} \leq S_t \leq S^{max} \qquad (1)$$

*where $S^{min}$ and $S^{max}$ are the minimum (14.8 billion m³) and maximum (74 billion m³) allowable reservoir storage.*

*Additionally,  $S^{begin}$  and  $S^{end}$  represent the initial and final reservoir storage (m³) for simulations (both of them are set as 65.1 billion m³), respectively, and are prescribed as:*

$$S_t = \begin{cases} S^{begin} & t = 1 \\ S^{end} & t = T \end{cases} \qquad (2)$$

*(c)  Reservoir release limits:*

*The reservoir release constraints are expressed as:*

$$QL_t \leq Q_t^{out} \leq QU_t \qquad (3)$$

*where  $QL_t$  and  $QU_t$  are the minimum and maximum release (m³/s) in period t, respectively. The expected guidelines for GERD reservoir water release are not explicitly available, thus releases are set higher than zero and lower than the maximum reservoir inflow (21.9 billion m³/month) during the high-flow season to reduce or eliminate downstream floods.*

*(d)  Power generation limits (Tesfa, 2013):*

$$PL_t \leq P_t \leq PU_t \qquad (4)$$

*where  $PL_t$  and  $PU_t$  are the minimum (0 MW) and maximum (6000 MW) power limits in period t, respectively.*

Regarding the upper power generation limit or installed capacity, it needs to be noted that there is uncertainty in media reports around the number for GERD which ranges from 5,150 MW (Ezega News, 2019a) to 6,000 MW (Ezega News, 2019b;Zelalem, 2020). The number (6,000 MW) which was used both in the annual electricity production estimation (aforementioned 15130GWh = 15130*1000/(24*365) = 1727 MW) and previous publicly available scientific studies (Tesfa, 2013;Yang et al., 2021) is opted in this study. It is worth noting that re-running the simulations with an installed capacity of 5,150 MW instead of 6,000 MW does not change principal conclusions. We have further clarified it in the revised manuscript as below.

*When completed, the GERD will become the largest hydroelectric dam (installed capacity more than 5,000 MW) in Africa (Government of Ethiopia, 2020) and will have a total reservoir capacity of 74 billion cubic meters. The GERD is expected to produce an average of 15,130 GWh of electricity annually (with mean output of 1727 MW) (Tan et al., 2017;Tesfa, 2013), which will contribute to Ethiopia's national energy grid and feed the East African power pool (Nile Basin Initiative, 2012). There is uncertainty in media reports regarding the total installed capacity for GERD which ranges from 5,150 MW (Ezega News, 2019a) to 6,000 MW (Ezega News, 2019b;Zelalem, 2020). A value of 6,000 MW, which was applied both in the annual electricity production estimation and previous*

*publicly available scientific studies (Tesfa, 2013;Yang et al., 2021), is opted for this study. It is worth noting that re-running the simulations with an installed capacity of 5,150 MW instead of 6,000 MW does not change principal conclusions.*

Ezega News: Power generation capacity of GERD slashed to 5150 MW—Ethiopian Minister, https://www.ezega.com/News/NewsDetails/7331/Power-Generation-Capacity-of-GERD-Slashed-to-5150MW-Ethiopian-Minister, 2019a.

Ezega News: Ethiopia dismisses reports of capacity reduction of GERD, https://www.ezega.com/News/NewsDetails/7321/Ethiopia-Dismisses-Reports-of-Capacity-Reduction-of-GERD, 2019b.

Tesfa, B.: Benefit of grand Ethiopian renaissance dam project (GERDP) for Sudan and Egypt, 2013.

Yang, G., Zaitchik, B., Badr, H., and Block, P.: A Bayesian adaptive reservoir operation framework incorporating streamflow non-stationarity, J Hydrol., 594, 125959, 2021.

Zelalem, Z.: Ethiopia and Egypt are pushing each other to the brink in a battle for control on the river Nile, Quartz Africa, https://qz.com/africa/1862962/ethiopia-egypt-battle-for-river-nile-grand-dam-without-trump/, 2020.

2. Following are the aspects that could be incorporated as recommendations for further work.

The study applies the historic sequence (from 1965 to 2017) just downstream of the GERD dam and reliability considerations are incorporated by the statistical treatment of the residuals of the linear function relating annual releases to annual inflows (equation 10) during low flow years. The resulting range of variability for the different exceedance levels as illustrated in Figure 4 is low and probably underestimates the impact of hydrologic variability. Since the historic sequence is not very long and seems to include only two severe drought periods (from 1978-1988 and 1991 - 1997 as seen on Figures 7 and also reflected on Figure 6d), the extension of the historic inflow records using (its correlation with) the longer-term records available in the Nile basin could enable a more realistic assessment of the effects of droughts on the system and how the GERD could be best operated during such severe droughts. It is during such periods of severe water shortage that tensions are likely to rise among the riparian countries. A more comprehensive probabilistic approach based on stochastically generated ensembles of the extended inflows could also be considered.

Reply: Thank you for your suggestions. We appreciate that you point out the impact of hydrological uncertainty on GERD operation simulations and optimizations. Yes, it is true that there are limited drought periods in the historical streamflow time series (during 1965-2017), which may lead to uncertainty in water sharing during severe droughts and may

underestimate the impact of hydrologic variability on GERD operation. That said, we did test our model by bootstrapping observations to create many sequences, including those having longer drought periods that historically observed, however the overall hydropower and release outcomes were virtually indistinguishable. Thus we opted to continue with the historical time-series.

However, we full agree with the reviewer, and for future work we will extend the streamflow record at the El Diem gauging station (which is located just downstream of GERD site) by relating it to other gauging stations in Nile basin. We agree that this synthetic GERD inflow records could further our understanding of the impacts of hydrological uncertainty on the trade-off between upstream benefits and downstream drought risks and the corresponding drought policy design.

In addition to the hydrological uncertainty, the uncertainty in drought policy design (e.g., choosing slope parameter $\alpha$ and exceedance parameter z in equation 10) should also be considered to better understand its influence on GERD power generation and downstream drought risk. Finally, the trade-off in objectives may be affected by land use or climate changes, and if significant, the drought policy may need to be adjusted accordingly.

We have included these future work opportunities in the conclusion of the revised manuscript as below.

*It is worth noting that there are limited drought periods in the historical streamflow time series, which may lead to greater uncertainty in water sharing during severe droughts and may result in underestimating the impact of hydrologic variability on GERD operations. To address this, the GERD inflow record could be extended by relating it to other long-term gauging stations in the Nile basin to capture more historical droughts and better characterize hydrologic conditions for enhanced policy design. In addition, the trade-off in objectives may be affected by land use or climate changes, and if significant, the drought policy may need to be adjusted accordingly in the future.*

---

## Author Response (AR2)

**Reply Letter**

**Title:** Water sharing policies conditioned on hydrologic variability to inform reservoir operations

**Author(s):** Guang Yang and Paul Block

**MS No.:** hess-2021-72 **MS type:** Research article

**Comments to the Author (pdf):**

1. Line 64-65: Remove brackets.

Reply: Thank you for your comments. The brackets have been removed.

2. Line 122: "mm" -> "mm/yr"?

Reply: Thank you for your comments. It has been corrected in the revised manuscript.

3. Line 125: "km3" -> "km3/yr"?

Reply: Thank you for your comments. It has been corrected in the revised manuscript.

4. Line 288, 302, 320, 321, 365: "BCM" -> "BCM/yr"?

Reply: Thank you for your comments. The "BCM" throughout this paper have been replaced with "BCM/yr" in the revised manuscript.

5. Line 317: "Comparing across rule types, rules with high mean output generate more hydropower mainly in wet years". Mean output of what?

Reply: Thank you for your comments. The mean output is the mean of power generation outputs (MW). It has been noted in the revised manuscript as below.

*Comparing across rule types, rules with high mean output of power generation generate more hydropower mainly in wet years*

6. Fig. 8: Is the Y axis referring to water release of inflow? Not clear from annual streamflow label.

Reply: Thank you for your comments. The Y axis refers to reservoir water release and inflow. We have changed the label of the Y axis to "Reservoir water release and inflow (BCM/yr)" in the revised manuscript.

**Non-public comments to the Author:**

1. Line 294-295 and 346 refer to Figures in Appendix S1 which is not included in the manuscript.

Reply: Thank you for your comments. All the figures in Appendix S1 were attached as "Supplement", they have been included in the section "Appendix S1: Supplemental Figures" of the revised manuscript.

2. The paper frequently gives volume of water in storage, and volume of water flowing. When giving volumes of water flowing, or fluxes the author should explicitly indicate this, e.g. 49 BCM/yr and not 49 BCM. Readers easily get confused with this.

Reply: Thank you for your suggestions. All volumes of water flowing throughout this paper have been explicitly stated, e.g., by replacing the "BCM" with "BCM/yr".

3. The above applies when stating annual rainfall, e.g. 1800 mm/yr and not 1800 mm

Reply: Thank you for your suggestions. This has been corrected in the revised manuscript.

4. The abbreviation BCM should be defined. This is not a commonly used and well known abbreviation

Reply: Thank you for your suggestions. The abbreviation "BCM" has been defined when it is first used in the revised manuscript as below.

*reservoir inflow values below the historical average (approximately 49 billion cubic meters per year, BCM/yr)*